# Comparison of primary and secondary particle formation from natural gas engine exhaust and of their volatility characteristics

Jenni Alanen[1], Pauli Simonen[1], Sanna Saarikoski[2], Hilkka Timonen[2], Oskari Kangasniemi[1], Erkka Saukko[1], Risto Hillamo[2], Kati Lehtoranta[3], Timo Murtonen[3], Hannu Vesala[3], Jorma Keskinen[1], and Topi Rönkkö[1]

[1]Aerosol Physics, Faculty of Natural Sciences, Tampere University of Technology, P.O. Box 692, 33101 Tampere, Finland
[2]Atmospheric Composition Research, Finnish Meteorological Institute, P.O. Box 503, 00101 Helsinki, Finland
[3]VTT Technical Research Centre of Finland Ltd., P.O. Box 1000, 02044 VTT, Espoo, Finland

*Correspondence to:* Topi Rönkkö (topi.ronkko@tut.fi)

**Abstract.** Natural gas usage in traffic and energy production sector is a growing trend worldwide; thus, an assessment of its effects on air quality, human health and climate is required. Engine exhaust is a source of primary particulate emissions and secondary aerosol precursors, which both contribute to air quality and can cause adverse health effects. Technologies, such as cleaner engines or fuels, that produce less primary and secondary aerosol could potentially significantly decrease atmospheric particle concentrations and their adverse effects. In this study, we used a potential aerosol mass (PAM) chamber to investigate the secondary aerosol formation potential of natural gas engine exhaust. The PAM chamber was used with a constant UV-light voltage, which resulted in relatively long equivalent atmospheric ages of 11 days at most. The studied retrofitted natural gas engine exhaust was observed to form secondary aerosol. The mass of the total aged particles, i.e., particle mass measured downstream of the PAM chamber, was 6-268 times as high as the mass of the emitted primary exhaust particles. The secondary organic aerosol (SOA) formation potential was measured to be 9-20 mg $kg_{fuel}^{-1}$. The total aged particles mainly consisted of organic matter, nitrate, sulfate and ammonium, with the fractions depending on exhaust after-treatment and the engine parameters used. Also the volatility, composition and concentration of the total aged particles were found to depend on the engine operating mode, catalyst temperature and catalyst type. For example, a high catalyst temperature promoted the formation of sulfate particles, whereas a low catalyst temperature promoted nitrate formation. However, in particular, the concentration of nitrate needed a long time to stabilize — more than half an hour — which complicated the conclusions but also indicates the sensitivity of nitrate measurements on experimental parameters such as emission source and system temperatures. Sulfate was measured to have the highest evaporation temperature, and nitrate had the lowest. The evaporation temperature of ammonium depended on the fractions of nitrate and sulfate in the particles. The average volatility of the total aged particles was measured to be lower than that of primary particles, indicating better stability of the aged natural gas engine emitted aerosol in the atmosphere. According to the results of this study, the exhaust of a natural gas engine equipped with a catalyst forms secondary aerosol when the atmospheric ages in a PAM chamber are several days. The secondary aerosol matter has different physical characteristics from those of primary particulate emissions.

# 1 Introduction

Primary aerosol particles are directly emitted into the atmosphere by various anthropogenic sources, such as vehicles, engines and power plants, and biogenic sources. Secondary aerosol particle mass forms as a consequence of the atmospheric oxidation of emitted precursor gases. In this process, the saturation vapor pressure of the organic and inorganic gases becomes lower, thus allowing them to transfer into particle phase through condensation and nucleation (Hallquist et al., 2009; Murphy et al., 2014). In addition to biogenic sources, also traffic and other anthropogenic sources contribute to secondary aerosol formation (Kanakidou et al., 2005).

Fine particles (<2.5 $\mu$ m) are found to cause adverse health effects and premature mortality in people (Dockery and Pope III, 1994; Lelieveld et al., 2015). The relative contribution of primary and secondary particles on these health effects is still unknown, but there are indications that secondary particles can be even more hazardous than primary particles (Künzi et al., 2015; McWhinney et al., 2011; Rager et al., 2011). Therefore, both primary and secondary particle emissions must be taken into consideration when evaluating the health effects of particle emissions.

Aerosols play an important role in the climate as well. Fine particles in the atmosphere affect the radiative balance of the atmosphere by either warming or cooling it, depending on their properties (Myhre et al., 2013); however, large uncertainties remain regarding the contribution of particles to climate change and its prevention. Clouds also contribute to the atmosphere's radiative balance. Aging of an aerosol can lead to increased hygroscopicity of the particles (Kanakidou et al., 2005) and a higher likelihood that they act as cloud condensation nuclei. The preservation and lifetime of the particles in the atmosphere partly define how large their impact is on the climate.

The formation process of secondary inorganic aerosol can be modeled rather accurately because the number of different inorganic precursors is small and their oxidation reactions are well known. Secondary organic aerosol (SOA) is a more complex subject area due to the vast number of different organic compounds, their potential reactions and the still unknown participation of all compounds in secondary aerosol formation (Hallquist et al., 2013; Jimenez et al., 2009). SOA has been a hot topic in aerosol science during the past decade (Huang et al., 2014; Robinson et al., 2007; Virtanen et al., 2010), but there are still many open questions in considering, for example, the SOA formation from vehicle emissions (Gentner et al., 2017). Also the relative fractions of secondary organic and inorganic aerosol from various emission sources still need to be studied. Both secondary organic and secondary inorganic aerosol can contribute significantly to air quality deterioration (Huang et al., 2014).

Particle number and mass emission regulations for passenger cars and heavy-duty engines have substantially decreased the primary particle emissions from vehicles (e.g., May et al., 2014; Johnson, 2009). Secondary particle precursor emissions or secondary aerosol formation potential are not directly regulated, but some of the current emission regulations affect secondary particle precursor emissions indirectly. For instance, oxidative catalysts reduce the total hydrocarbon emissions and thus probably the emissions of secondary organic aerosol precursors; simultaneously, they also change the oxidation state of inorganic compounds. Furthermore, the mandatory national targets of 10 % biofuel (ethanol) in gasoline in EU may have decreased the SOA formation in the atmosphere (Timonen et al., 2017). In general, vehicles emit a substantial fraction of anthropogenic precursors for SOA formation (Gentner et al., 2017), and the amount of potential SOA often exceeds the emissions of primary

organic aerosol. For instance, gasoline vehicles emit 9-15 times or even two orders of magnitude higher secondary organic particulate matter than primary organic particle mass (Karjalainen et al., 2016; Nordin et al., 2013; Platt et al., 2013; Tkacic et al., 2014). Indeed, Bahreini et al. (2012) found that secondary organic aerosol originating from gasoline engines forms the majority of the SOA in and downwind of large metropolitan areas. From diesel vehicles without a particle filter, the SOA mass formation potential is of the same magnitude as or lower than the primary particle mass emission (Jathar et al., 2017; Gordon et al., 2014b; Weitkamp et al., 2007).

Exhaust after-treatment can reduce secondary aerosol formation potential from engine exhaust, especially SOA formation potential. In general, diesel fuel has the strongest secondary organic aerosol formation potential amongst diesel, jet fuel, gasoline and Fischer-Tropsch from natural gas and coal (Jathar et al., 2013). However, diesel vehicles equipped with oxidation catalysts or catalytic particle filters have been reported to be minor secondary particle emitters (Chirico et al., 2010; Gordon et al., 2014b; Samy and Zielinska, 2010). In gasoline engine functioning exhaust after-treatment can also clearly reduce secondary particle formation (Karjalainen et al., 2016). The secondary aerosol precursor emissions of engines and vehicles are also strongly dependent on the driving conditions, which should be taken into account in emission comparisons.

For instance, Tkacic et al. (2014) showed that the secondary inorganic mass often exceeds the amount of the secondary organic aerosol in a highway tunnel, even by a factor of two. The main contributor to secondary inorganic aerosol in their study was ammonium nitrate, which originates from $NO_x$ and ammonia emissions. According to Karjalainen et al. (2016), large fractions of nitrate in the secondary inorganic particles are characteristic for highway driving, and the inorganic species concentrations are relatively low when compared with the secondary organic aerosol formed during other parts of the New European Driving Cycle (NEDC), which they tested. Idling is another engine operation mode that can produce significant amounts of secondary inorganic aerosol from gasoline vehicle exhaust (Nordin et al., 2013).

Natural gas usage as a fuel in combustion engines, both in energy production and traffic, is a growing trend worldwide. Natural gas engines emit little primary particle mass and less $CO_2$ than engines fueled with conventional fuels (Anderson et al., 2015; Bielaczyc et al., 2014) but their particle number emission can be significant (Hallquist et al., 2013; Jayaratne et al., 2010). In addition, the size of the majority of the particles emitted by natural gas engines can be below the detection limits of traditional exhaust particle measurement devices (Alanen et al., 2015). Natural gas engine exhaust particles are highly volatile (Bullock and Olfert, 2014; Jayaratne et al., 2012) or they can consist of volatile matter condensed on non-volatile cores (Alanen et al., 2015; Graves et al., 2015; Pirjola et al., 2016). The evaporation of the particles is largest at temperatures below 100 °C (Alanen et al., 2015; Jayaratne et al., 2012). Primary particles from natural gas engines mainly consist of organic matter (Pirjola et al., 2016), but the composition depends on exhaust after-treatment (Lehtoranta et al., 2016). In the study of Lehtoranta et al. (2016), high catalyst temperatures were found to increase the fraction of sulfate in particles when a combination of oxidative and reductive catalysts was employed. Also, increased ammonium concentrations were found in particles at high catalyst temperatures.

To the authors' knowledge, there are no published studies on secondary particle formation from natural gas engine emissions, its chemical or physical properties or the effect of exhaust after-treatment on exhaust's secondary particle formation. Goyal and Sidhartha (2003) recorded a notable improvement in the air quality of Delhi when a portion of vehicles were changed

to natural gas vehicles in 2001. In our study, the secondary aerosol formation potential of natural gas engine exhaust was investigated using a flow-through reactor, and the chemical and physical characteristics of particles were investigated by aerosol instruments. The results were compared to those of primary particle emissions, but because the primary particle emissions of the same engine have already been discussed in two earlier publications (Alanen et al., 2015; Lehtoranta et al., 2016), they are not a focus of this paper. In general, the aim of this study is to report the total particulate emissions of natural gas engines, i.e., primary and secondary particles, to ensure that shifting to natural gas from diesel and gasoline will not cause unexpected environmental or health issues, and to define the possible benefits of the shift. Volatility studies on both primary and secondary particles enabled an evaluation of the stability and residence time of the particles in the atmosphere. The study of chemical composition can help solve their origin and find ways to reduce the particulate emissions.

## 2  Methods

### 2.1  Engine and after-treatment

A small (2.0 liter displacement) spark ignited passenger car engine was used for the measurements with Russian pipeline natural gas as fuel. The methane content of the fuel was 97 %, other hydrocarbon content was 1.6 % and nitrogen content was 0.9 %. The sulfur content was below 1 ppm. The engine was run at two steady-state engine operation modes with torque of 70 Nm and speed of 2700 rpm (Mode 1, M1) and torque of 35 Nm and speed of 3100 rpm (Mode 2, M2). In engine mode 2, short chain hydrocarbons were added into the exhaust to make it resemble the exhaust of a power plant NG engine. The exhaust gas composition in two operation modes simulated typical natural gas power plant exhaust gas composition. The engine, natural gas and lubricating oil properties as well as the engine operation modes have been described in more detail by Murtonen et al. (2016), Alanen et al. (2015) and Lehtoranta et al. (2016).

Two separate after-treatment systems were applied in the measurements, both consisting of a reductive and an oxidative section. The after-treatment has been described in more detail by Lehtoranta et al. (2016). The first catalyst (Catalyst 1, C1) consisted of only one reactor, which targeted both oxidation of carbon compounds and $NO_x$ reduction through urea injection in the same catalyst reactor. The second catalyst system (Catalyst 2, C2) consisted of a palladium and platinum containing methane oxidation catalyst followed by urea injection and a vanadium-SCR catalyst, which catalysts were supported on metallic honeycomb substrates. Catalyst 1 was used in three different exhaust temperatures in the range of 350-450 °C in order to study its performance and its influence on secondary particle formation potential of the engine exhaust. The temperature of the catalyst 2 was 500 °C. Catalyst performance depends on the exhaust temperature (e.g., Lehtoranta et al., 2016). By using the catalysts at different temperatures, effects of catalyst temperature on the formation and characteristics of primary and total aged particulate matter could be studied. The catalyst temperatures were measured upstream of the oxidation catalysts. The temperature prior to the SCR of catalyst 2 was approximately 50 °C lower than prior to the oxidation catalyst. The exhaust flow through the catalysts was kept constant at 80 kg h$^{-1}$ by leading only a part of the exhaust gas flow through them (Murtonen et al., 2016).

## 2.2 Instrumentation and data analysis

The sampling system or the particle measurement instruments consisted of a porous tube diluter (PTD, Mikkanen et al. 2001; Ntziachristos et al. 2004) with a dilution ratio (DR) of 6, followed by a residence time chamber with a residence time of 6 s. The low primary dilution ratio was used because of the very low primary particle emission concentrations (See Alanen et al., 2015). The dilution air was heated to 30 °C to achieve constant dilution conditions. A second dilution stage was carried out with an ejector diluter (Dekati Ltd.) with DR 4. The dilution ratio over the PTD was adjusted using a bypass flow mass flow controller placed downstream the residence time chamber (Fig. 1). The dilution ratios were calculated from $CO_2$ concentrations in the raw and diluted exhaust samples and they could be used to calculate tailpipe concentrations of particle emissions. The aerosol sampling was done downstream the exhaust after-treatment system.

A potential aerosol mass chamber (Kang et al., 2007, 2011; Lambe et al., 2011, 2015) was used to simulate the aging process of an aerosol in the atmosphere. In the PAM, an oxidative environment ($O_3$, OH and $HO_2$, UV-light) was produced using two UV lamps emitting 185 nm and 254 nm radiation, respectively, in a small (13 l) flow-through chamber. The PAM chamber was placed between the two dilution stages, and the flow through it was a constant 5 liters per minute (residence time 156 s), measured by a bubble flow meter (Gilibrator, Sensidyne Inc.) and adjusted by a pressure regulator of the compressed air flow to the ejector diluter. The PAM chamber could be either bypassed or used to measure the properties of primary and secondary aerosols, respectively.

The approximate atmospheric age, i.e., the photochemical age simulated by the PAM chamber UV lights, was modeled using the properties of the PAM chamber and the measured concentrations of gaseous components that cause external OH reactivity in the chamber. The model used for calculating the OH exposures was based on the degradation mechanism extracted from the Master Chemical Mechanism or MCM v3.3.1 (Jenkin et al., 1997, 2003; Saunders et al., 2003) via the website http://mcm.leeds.ac.uk/MCM and translated to Matlab code using Kinetic PreProcessor or KPP (Damian et al., 2002). The model has been tested against $SO_2$ reduction measurements in the PAM chamber.

MCM is a near-explicit chemical mechanism that describes the atmospheric degradation of volatile organic compounds in gas phase. MCM describes the degradation of a given VOC through different generations of products until $CO_2$ is ultimately formed. It contains about 17000 reactions for 6700 different species. (Jenkin et al., 1997, 2003; Saunders et al., 2003). To be able to use these mechanisms with the PAM chamber, the photolysis rates have been calculated for ultraviolet light with wavelengths of 185 nm and 254 nm. The absorption cross section and quantum yield values needed for this are IUPAC recommendations (Atkinson et al., 2007) supplemented with the JPL data evaluation number 18 (Burkholder et al., 2015), when necessary. Some photolysis reactions relevant that are relevant to the PAM chamber but missing from the tropospheric MCM schemes have also been added.

KPP is a software tool for translating kinetic chemical mechanisms into Fortran77, Fortran90, C or Matlab simulation code. The generated code produces concentrations of each species present as a function of time (Damian et al., 2002). For the model used, the KPP source code was modified to fix certain conflicts involving the MCM mechanism and the photolysis rate calculations written for the PAM chamber as well as to allow large chemical schemes typical to MCM.

In this paper, we describe the PAM OH exposure as photochemical age, which is the equivalent time in the atmosphere in which the sample would reach the same OH exposure as in the PAM chamber. Thus,

$$\text{Photochemical age (days)} = \frac{\text{OH exposure}}{1.5 * 10^6 \text{ molec. cm}^{-3}} \frac{1}{3600 \text{ h}^{-1}\text{s} * 24 \text{ d}^{-1}\text{h}}, \tag{1}$$

where $1.5 * 10^6$ molec. cm$^{-3}$ is the average OH concentration in the atmosphere (Mao et al., 2009).

Relative humidity (RH) was measured downstream of the PAM chamber. The RH was high, about 80 %, due to the low primary dilution ratio that was applied during the experiments. The high RH of the sample complicated the evaluation of the PAM background mass levels — i.e., the particulate mass that was generated by only compressed air and UV lights — because the high RH could not be reproduced in the compressed air by the available instrumentation. The background levels were measured using both dry compressed air and compressed air with RH ~30 %.

NO$_x$ concentration was measured with a chemiluminescence detector (CLD), CO and CO$_2$ concentrations in raw exhaust were measured with a non-dispersive infrared (NDIR) analyzer and CO$_2$ concentrations in diluted exhaust were measured with a Sick Maihak SIDOR gas analyzer. Water, methane, NH$_3$, HNCO and the ratio of NO and NO$_2$ were measured with a Fourier transform infrared spectroscopy (FTIR, Gasmet Cr-2000) analyzer, and methane, ethane, propane and ethylene were measured with a gas chromatograph (GC).

Aerosol instruments covered a large range of particle mobility and aerodynamic size as well as measurements of the particles' chemical composition. An engine exhaust particle sizer (EEPS, TSI Inc., Mirme 1994) and a high-resolution low-pressure impactor (HRLPI, Arffman et al. 2014) were employed, both on one-second time resolution, to measure the particle number, mass and size. The EEPS measures the size distribution and concentration of particles with a mobility diameter of 5.6-560 nm, and the HRLPI measures the aerodynamic size distribution of particles with diameter of ~5-200 nm. The mass of the particles

was calculated under assumptions of unit density and spherical particles. EEPS default inversion was applied.

A soot particle aerosol mass spectrometer (SP-AMS, Aerodyne Research Inc., US) — a combination of a high-resolution time-of-flight aerosol mass spectrometer and a single particle soot photometer (Droplet Measurement Technologies) — was used to measure the chemical composition and oxidative state of the aerosol sample. The SP-AMS measures both refractory and non-refractory particulate matter. It operated in V-mode with a one-minute time resolution, measuring half of the time in

mass spectra (MS) mode and the other half in particle size (pToF) mode. Both laser and tungsten vaporizers were used. The collection efficiency applied in the calculations was calculated using the the parametrization by Middlebrook et al. (2012). The CO$_2$ gas interference in the AMS data was corrected by using the CO$_2$ concentrations measured with Sick Maihak SIDOR gas analyzer. The impact of ammonium nitrate interference on CO$_2^+$, O/C and H/C ratios was evaluated to be small (less than 5 % for O/C and H/C; Pieber et al. 2016). Therefore, a correction of ammonium nitrate interference was not applied for organics or

O/C data.

Volatility measurements were made with a thermodenuder described in the publication by Heikkilä et al. (2009). When the remaining mass of particles was measured as a function of TD temperature, the thermodenuder was heated up to 265 °C and

then switched off, with the sample flow still flowing through it. The decreasing temperature was recorded for at least half an hour until the temperature was below 50 °C.

Emission factors were calculated from fuel composition and engine performance information. Residual $O_2$ in the exhaust was 6.2-6.3 %, the power of the engine was 12 kW and 20 kW and the combustion air flow into the engine was approximately 100 and 115 kg h$^{-1}$ in engine modes 1 and 2, respectively. Calculated from the fuel composition information, the emission factor for $CO_2$ $EF_{CO_2}$ was 2730 g kg$^{-1}_{fuel}$, and the carbon intensity was 0.74 kg$_C$ kg$^{-1}_{fuel}$.

## 3 Results and discussion

### 3.1 Secondary particle formation and chemical composition

The concept "total aged" here comprises all particle mass measured downstream of the PAM chamber, i.e., both primary and secondary particle mass. In general, primary particle mass has not been subtracted from the mass measured downstream of the PAM (total aged) to calculate the secondary particle mass separately because doing so would have created inconsistency in representation of the results, since, for example, particle size distributions or volatility behavior cannot be presented in this way. For the same reasons, the PAM background mass — i.e., the particle mass generated in the PAM chamber from clean compressed air — has not been subtracted but is instead presented separately in the supplementary section of this paper. To enable a comparison to literature, an exception is made when presenting secondary particle production factors.

Figure 2 contains particulate mass measurement results derived from the three aerosol instruments. The chemical compositions from SP-AMS are also presented. The cases (engine mode, catalyst and catalyst temperature) included in this paper cover all of the tested exhaust temperatures and both engine operation modes, and they have data collected with all available instruments of both the primary and total aged aerosol measurements. In most cases in our measurements, primary exhaust particle mass concentrations from the natural gas engine were close to the detection limits of the instruments EEPS, HRLPI and SP-AMS (Alanen et al., 2015; Lehtoranta et al., 2016). Exceptions were made by the high temperature catalyst cases (M2, C2, 500 °C and M2, C1, 450 °C) during which more primary particle mass was formed, especially on the size ranges of the HRLPI and EEPS: a high catalyst temperature favors the conversion of $SO_2$ into $SO_3$ and further into sulfuric acid, which can nucleate and condense on existing particles in the sampling process or when released into the atmosphere (see, e.g., Arnold et al., 2012; Rönkkö et al., 2013). The primary particle formation phenomena and concentrations have been discussed in more detail in Lehtoranta et al. (2016) while this paper focuses on secondary aerosol formation and the total aged particle emissions.

The formed secondary particle mass concentrations were found to be significantly high in comparison with the primary particle mass emissions. In all of the investigated cases, particle mass increased when the sample was led through the PAM chamber. The increase in mass could be magnitudes larger than the primary particle mass emission (Table 1). The relative increase in mass in the PAM chamber could not be specified for all of the HRLPI measurements because of the very low primary particle mass. The total aged aerosol mass produced by natural gas engine exhaust was 0.99-2.6 mg m$^{-3}$ according to SP-AMS, leading to secondary mass production of 0.96-2.5 mg m$^{-3}$.

The secondary aerosol formation — i.e., the ratio of the total aged particulate mass to the primary particulate mass — was lower in the cases that already produced more primary particle mass, i.e., in the cases with a high catalyst temperature. It is possible that if the catalyst conditions are favorable, the particulate matter that would otherwise condense on particles in the PAM chamber condenses on particle phase already in the cooling and dilution process. In other words, if the catalyst sufficiently oxidizes the exhaust gases, thus lowering their saturation vapor pressure, they condense or nucleate already when released from the tailpipe and not later on in the atmosphere. A high catalyst temperature promoted larger total aged aerosol formation, according to the EEPS and HRLPI measurements. However, the total aged mass concentrations of the SP-AMS did not increase as catalyst temperature increased. The differences in instrument's showings are discussed in Sect. 3.3. Also, the variation in the atmospheric ages increases uncertainty in the comparison of the catalyst temperature on secondary aerosol formation potential.

The total aged aerosol of the natural gas engine exhaust consisted of both organic and inorganic matter at the tested operating conditions (Fig. 2, Table 1). Approximately half of the total aged aerosol particle mass detected by SP-AMS consisted of organic matter. The fraction of sulfate and nitrate was measured to be 34-49 % in total, with their ratio depending on the case, and the fraction of ammonium varied between 10-15 %. Link et al. (2017) found that even high $NO_x$ emissions can produce negligible amounts of secondary nitrate aerosol if related ammonia emissions are small. Because secondary ammonium nitrate aerosol formation is limited by ammonia, its formation is probably more related to the exhaust after-treatment than the fuel. The exact ammonia concentrations in the raw exhaust cannot be given because they were below the instrument detection limit 2 ppm. According to these measurements, also low ammonia emissions may have atmospheric importance as secondary inorganic aerosol precursors.

The organic fraction of the total aged aerosol consisted of hydrocarbon fragments ($C_xH_y$), fragments with one oxygen atom ($C_xH_yO$) and fragments with more than one oxygen atom ($C_xH_yO_{z,z>1}$); there was little or no $C_xH_yN$ fragments (hydrocarbons with nitrogen) in the total aged particles. The main secondary organic ions detected by the SP-AMS were $CO_2^+$, $CHO^+$ and $C_2H_3O^+$. The composition of the organic aerosol was similar in all of the cases: the $C_xH_yO_{z,z>1}$ group was the largest, followed by $C_xH_y$ and $C_xH_yO$. The source of the secondary organic aerosol could be either the natural gas or the lubricating oil. However, we are not able to tell the source based on these measurements. The fuel mainly consisted of light hydrocarbons that are unable to form secondary organic aerosol (e.g., Seinfeld and Pandis, 2016, pp. 575). For example, Thiruvengadam et al. (2014); Eichler et al. (2017) have suspected engine lubricating oil to be responsible for a large portion of engine emitted particles. Therefore, we believe that also lubricating oil is a potential candidate for the source of secondary aerosol.

The O/C ratios of the total aged aerosol measured by SP-AMS were between 0.9 and 1.2. The O/C ratio of the primary aerosol in the case with the largest concentration was slightly smaller (1.1) than the O/C ratio of the total aged aerosol in the same case (M2, C2, 500 °C). In all of the other primary aerosol measurements, the particle mass concentrations in the sample were too low for O/C ratio analysis. In comparison with a secondary aerosol emission study on gasoline engines by Karjalainen et al. (2016), the observed O/C ratios in the total aged aerosol from the PAM chamber were rather high.

The emission factors or secondary aerosol production factors in different units can be calculated from the presented particle mass concentrations by using the following factors. If a unit factor mg $kg_{fuel}^{-1}$ is needed, a factor of ca. 22 $m^3$ $kg_{fuel}^{-1}$ can be applied to multiply the particle concentration (Calculation e.g. in Jathar et al., 2017; Gordon et al., 2014b). In order to obtain emission and production factors in unit $kWh^{-1}$, a factor 2.7 $m^3$ $kWh^{-1}$ (Mode 1) or 4 $m^3$ $kWh^{-1}$ (Mode 2) can similarly be used. These factors are derived from the fuel composition and engine performance information provided in Sect. 2.1 and 2.2 and the exhaust $CO_2$ concentration.

The production factors of secondary organic aerosol have been calculated and collected in Table 2, in unit $kg_{fuel}^{-1}$. To be able to compare the SOA production factors, here primary organic aerosol was subtracted from the total aged organic aerosol. Table 2 also contains SOA production factors of secondary organic aerosol for different diesel and gasoline vehicles obtained from the literature. Although the total aged particulate matter production of the investigated NG engine was much larger than its primary particle emissions, it was smaller than SOA production from in-use diesel and gasoline vehicles in the literature (Tkacik et al., 2014). The SOA formation potential from the NG engine, measured by SP-AMS, was similar to that of a diesel vehicle equipped with a catalytic converter or to that of a hot start gasoline vehicle. On the other hand, the photochemical age that was simulated by a chamber in the different studies varied greatly. This is why the comparison of the SOA production factors should be done very carefully, if at all. The longest atmospheric ages in the literature collected in Table 2 were achieved in our study.

Palm et al. (2016); Tkacik et al. (2014); Kang et al. (2011) have seen with an oxidation flow reactor — such as the PAM chamber in our experiments — the highest potential secondary organic aerosol formation takes place at photochemical age of a few days and starts decreasing after that. For example, in the vehicle fleet emission study in a highway tunnel of Tkacik et al. (2014), the peak secondary aerosol production took place after 4-10 days of equivalent atmospheric oxidation ([OH]=1.5*$10^6$ molec. $cm^{-3}$), and larger OH exposures started to reduce the secondary mass. In our study, the simulated atmospheric, or photochemical, ages in the investigated cases varied between 4.6 and 10.7 days, depending on the external OH reactivity, which was affected by the concentrations of gaseous emissions (Table 1) entering the PAM chamber and by the relative humidity of the sample. The largest total aged particle concentrations were achieved with the longest atmospheric ages and the lowest particle concentrations were achieved with the shortest atmospheric ages. However, the secondary aerosol formation potential may have also been affected by the engine parameters and not only the achieved photochemical age: the total aged particle concentrations were the highest in engine operation mode 2 (M2).

## 3.2 Volatility of primary and secondary particle mass

The volatility of the particles was studied with a thermodenuder. Mass fraction remaining (MFR) stands for the fraction of the particle mass at a given thermodenuder temperature and the particle mass at room temperature. In Fig. 3, the particle mass fraction remaining has been calculated for two representative cases of primary emissions and four representative cases of total aged particle emissions, selected from among the cases already introduced. Figure 3 only shows data from EEPS, since the curves obtained from HRLPI were similar. Here, the curves have been smoothed by a moving average but the original one-second-resolution figure can be found in the supplementary section. For the total aged emissions, the cases with both higher

and lower catalyst temperature are presented. For primary particle emissions, only the case with the higher catalyst temperature is presented. This is because an accurate examination of the volatility of primary particles in low catalyst temperatures could not be done, due to the insufficient primary particle mass concentrations for high-quality analysis. In the figure, the "starting point", i.e. the temperature where the mass fraction remaining is one, is 50 °C and not lower because of the decelerated cooling
of the thermodenuder toward the room temperature and related time limitations.

The MFR curves for each type of particles are characteristic, i.e., each particle type can easily be distinguished by their evaporation behavior. To highlight this, the primary particle evaporation is marked with black, and the total aged particle evaporation curves are marked with cyan and blue in Fig. 3. The volatility of the particles from the natural gas engine clearly changed when the particles were aged. At high catalyst temperature, the primary particles (black triangles in Fig. 3) were
more volatile than the total aged particles (cyan squares). Approximately half (46-60 % in EEPS, 43-53 % in HRLPI) of the total aged particle mass remained at a thermodenuder temperature of 250 °C, whereas only 5-10 % (1-4 % in HRLPI) of the primary particle mass remained at that temperature. Also the catalyst temperature had an impact on the volatility of the total aged particles (blue vs. cyan). An easily evaporable fraction of the total aged particles was formed in the case of a low catalyst temperature, which evaporated below 110 °C. Because of this easily evaporable fraction, the MFR of total aged particles at
250 °C was 30 % in the low catalyst temperature cases, while in the high catalyst temperature cases the MFR of total aged particles at 250 °C was 46-60 %.

The thermodenuder used in this study has been designed to minimize nanoparticle losses by reducing the residence time (Heikkilä et al., 2009). For example, in this study, the residence time in the heated zone of the thermodenuder was less than one second. An et al. (2007) measured the volatility of secondary organic aerosol produced during $\alpha$-pinene photo-oxidation
and observed that only half of the secondary particle mass evaporates in a thermodenuder (100 °C) if the residence times in the heated zone of the thermodenuder are less than a few seconds. With longer residence times, the remaining mass downstream of the thermodenuder decreases to less than three percent. This means that the remaining fraction of particle mass in our study could have been smaller with longer residence times in the thermodenuder. On the other hand, a longer residence time in the thermodenuder would have increased the nanoparticle losses. In this study, with the use of a thermodenuder, we could observe
the volatility differences between the different types of particle emissions emitted by a natural gas engine.

In Fig. 4 the remaining mass fractions are plotted for different chemical species of the particles drawn from the SP-AMS. In the primary emission case, approximately one third of the particle mass — consisting mainly of organics — remained at TD temperature of 250 °C. The low concentrations and particle size below the detection limit of SP-AMS degrade the analysis in the case M2, C1, 450 °C, which was seen as a fluctuating signal. About 25 % of the total aged particle mass in the high catalyst
temperature cases and less than 10 % of the total aged particle mass in low catalyst temperature cases remained at 250 °C, according to SP-AMS. The remaining particle matter consisted of organics, sulfate and ammonium, in this order.

The composition information reveals that the high-volatility fraction of the total aged particles in the low temperature catalyst cases consisted of nitrates, possibly of ammonium nitrate, and high-volatility organics. The primary particle sulfate evaporated at thermodenuder temperatures between 100-170 °C, and the total aged particle sulfate evaporated more gradually above 120
35   °C. In all types of particles, the evaporation of organics was steady and gradual below 200 °C, indicating various organic

compounds with different evaporation temperatures. Above 200 °C, the evaporation of organics decreased. This combined SP-AMS and EEPS/HRLPI derived thermodenuder temperature ramp information can be used in future measurements for particle composition analysis: The evaporation temperature of the particles can give valuable information about the composition of the particles also without an access to SP-AMS.

The temperatures at which 50 % of the volatile fraction of the chemical compounds of the particles were remaining are collected in Table 3. The case "Primary, M2, C1, 450 °C" had particle mass concentrations that were too low (See Fig. 4) for this kind of examination. The evaporation temperatures of sulfate and nitrate were the highest and the lowest, respectively, in all of the analyzed cases (all catalyst temperatures; primary and total aged particles). Similarly to Huffman et al. (2009), who measured ambient aerosol volatility in megacities with a thermodenuder and an SP-AMS, we found that nitrate had the highest volatility and sulfate had the lowest.

Robinson et al. (2007) and Huffman et al. (2009) proposed that all organic aerosol should be considered semivolatile. Our results on primary and PAM chamber generated organic aerosols point in that direction as well. The evaporation temperature of the volatile fraction ($T_{volatile,\ 50\ \%}$) of organic matter lay between the $T_{volatile,\ 50\ \%}$ of nitrate and sulfate in all cases. Also, a significant fraction of the mass concentration of the organic matter did not evaporate. More exact specifications of the volatility cannot be given, but there is room left for speculation if part of the organic matter in secondary particles is SV-SOA or LV-SOA (Murphy et al., 2014).

The ammonium in total aged particles evaporated at higher thermodenuder temperatures when the catalyst temperature was high. The theory that the sulfate-nitrate trade-off phenomenon that determines the formation of nitrates is ammonium-bound is supported by the evaporation temperatures of ammonium. Ammonium evaporated at approximately 20 °C higher thermodenuder temperatures in the high catalyst temperature cases (Table 3); thus, its evaporation temperature was closer to the evaporation temperature of sulfate when the sulfate concentration of the particles was larger. By contrast, in the low catalyst temperature cases where the nitrate concentration was higher, the evaporation temperature of ammonium was closer to that of nitrate.

The nitrate concentrations measured during the thermodenuder temperature ramp (Fig. 4) in low thermodenuder temperatures differed from the nitrate concentrations that were measured without a thermodenuder (Fig. 2 a+b) in total aged particles. A possible explanation is that a long time is needed for the nitrate concentration to stabilize. In our measurement protocol, we waited 10-15 minutes after switching the PAM UV-lights on, followed by a 10-minute steady-state measurement with the aerosol instruments. After this, a thermodenuder ramp was started, which took approximately 45 minutes. Based on the results, the 10-15-minute wait was insufficient if accurate nitrate concentrations were desired. Therefore, the chemical compound measurements performed at low thermodenuder temperature can give a truer picture of the secondary aerosol formation than the measurements presented in Fig. 2. The change in concentrations between the steady-state measurements and the thermodenuder ramp measurements was the largest for nitrate, but the concentrations of other compounds also differed slightly from each other. Because the nitrate concentrations were found to be the slowest to stabilize and the most sensitive to changes in the system, such as to changes in temperature, special attention should be given to measurements of nitrate, especially when a

PAM chamber is being used. We note that because nitrate formation is limited by ammonium, the slow stabilization is probably related to ammonia.

According to our thermodenuder temperature ramp experiments, the catalyst temperature affected the total aged particle composition. With a decreasing catalyst temperature, the mass concentration and fraction of sulfate in total aged particles de-
creased (Fig. 4, Table 1). This was expected: at lower catalyst temperatures the oxidation of $SO_2$ to $SO_3$ decreases and less sulfuric acid (sulfates) can form (Arnold et al., 2012). The mass concentration of nitrate in secondary particles increased as the catalyst temperature decreased. This could not be explained by catalyst performance improvement: gaseous $NO_x$ levels remained similar at all catalyst temperatures or rose as catalyst temperature increased (see Lehtoranta et al., 2016). Because ammonia concentrations after catalyst were low, below 2 ppm in all cases, the effect of catalyst temperature on ammonia emis-
sion could not be measured. However, ammonium concentrations measured by SP-AMS correlated rather well with nitrate concentrations. Therefore, we suggest that ammonium increase was related to the nitrate increase. Also the sulfate concentrations could partly explain the behavior of the nitrate concentrations. If enough gaseous sulfuric acid is available, ammonium sulfate forms, and if not, more ammonium nitrate can form instead. Similar behavior of nitrate and sulfate trade-off has been measured by Ntziachristos et al. (2016) for two different marine fuels, namely heavy fuel oil (HFO) and light fuel oil (LFO).

### 3.3   Differences between instruments and mass size distributions

Slightly unexpectedly, the total aged particle mass measured by SP-AMS was 2-4 times larger than the total mass measured by EEPS and 1-3 times larger than that measured by HRLPI (Fig. 2). There could be several reasons for this. In EEPS and HRLPI, unit density and spherical particles were assumed in the mass calculations. Natural gas engine primary particles have a density of 0.85 g cm$^{-3}$ (Bullock and Olfert, 2014), but the densities of natural gas engine secondary particles can be larger
than the unit density. For example, the density of ammonium nitrate, ammonium sulfate and sulfuric acid is approximately 1.5, 1.5 and 1.8 g m$^{-3}$, respectively (Clegg and Wexler, 2011). Particle density does not completely explain the difference in instrument readings. Also, the collection efficiency (CE) estimation used in SP-AMS calculation is probably not the reason for the differences between the instrument results in this study. Evaluation of the CE following the procedure of Middlebrook et al. (2012) revealed that CE = 0.45 was the correct value for the studied total aged particles in the cases in Figure 1.
However, the detection efficiency and size range varied among the aerosol instruments (EEPS 5.6-560 nm, HRLPI ~5-200 nm, SP-AMS ~30-1000 nm) and can explain the differences in results; HRLPI can detect a larger fraction of the primary particles than SP-AMS because of the more suitable size range of the instrument and, correspondingly, SP-AMS can detect a larger fraction of the total aged particles formed in the PAM chamber because of its more suitable size range. Also, particle losses may play a role in the differences between instruments; particle losses in the PAM chamber were larger in the HRLPI size
range than in the SP-AMS size range. Nevertheless, most probably, the largest role was played by the differences in instrument size ranges.

Mass size distributions of the total aged aerosol, measured with EEPS and HRLPI, are plotted in Fig. 5. HRLPI suggests that a part of the particle mass lies above the instrument size range, which was confirmed by SP-AMS mass size distributions in Fig. 6. According to SP-AMS, the mass size distributions of total aged particles were bimodal, with the size of the larger mode being

480-840 nm and the smaller being 150-200 nm. The mode with smaller particle size was dominated by organics. Although the mass concentration of the total aged particles was better recorded by SP-AMS, a portion of the particles on the smallest particle sizes was missed due to the lower limit of SP-AMS size range at 30-50 nm. The best overall picture of is therefore gained with a combination of SP-AMS and HRLPI. See supplement for a comparison of the size distributions measured by different instruments in the same figure. The two instruments that measure the aerodynamic diameter of the particles (HRLPI and SP-AMS) compare quite well with each other in the size range 47-124 nm.

We can also see a difference between the EEPS and HRLPI mass size distributions. The difference is probably due to the inversion of EEPS, which forces the size distributions to follow a log-normal shape. EEPS also underestimated the mass of particles with diameter above 200 nm (see supplement). The different measurement principles of the instruments must also be kept in mind. EEPS measures the mobility size and HRLPI measures the aerodynamic size of the particles.

### 3.4 PAM artifacts and losses

The so called smog chambers are an established method of measuring SOA formation. An oxidation flow chamber such as PAM provides some advantages in comparison to smog chambers, such as a higher degree of oxidation, smaller physical size and a short residence time, which allows measurements with higher time resolution. Smog chamber walls may also cause large wall losses and influence the chemistry in the chamber (Bruns et al., 2015). On the other hand, smog chambers may simulate the atmospheric oxidation of organic precursors better than oxidation flow chambers due to their more tropospheric oxidant concentrations and longer residence times (Lambe et al., 2011).

The PAM method has been designed to produce the maximum potential aerosol mass from precursor gases (Kang et al., 2007). In that stage, the oxidation products of precursors have condensed into the particle phase and formed secondary aerosol. However, because the oxidant concentrations are unrealistically high in PAM, the UV light intensity used is non-tropospheric and the residence times are much shorter than in atmosphere (e.g., Simonen et al., 2017), precursor oxidation products have also other possible fates; they can be oxidized too far and form non-condensable oxidation products before condensation (accelerated chemistry) and they can exit the reactor before the condensation occurs. Also, precursor oxidation products can be lost on the PAM walls although the losses on the walls are minimized by the chamber design (Lambe et al., 2011). These other fates than condensing on particle phase are viewed here as PAM artifacts and losses.

The losses of condensable organic oxidation products and artifact effects of the accelerated chemistry in the PAM have been evaluated following the method of Palm et al. (2016) for the cases in Figure 2. HRLPI number size distributions were used to calculate the condensation sink needed in the loss calculation. Molar mass of 200 g mol$^{-1}$, diffusion coefficient of $7*10^{-6}$ m$^2$ s$^{-1}$ (Tang et al., 2015) and rate constant for reaction with OH of $1*10^{-11}$ (Ziemann and Atkinson, 2012) were applied. For sticking coefficient selection $\alpha = 1$ (assumed by Palm et al. 2016), the fraction of oxidation products condensed into the particle phase was $0.94 \pm 0.03$, but for $\alpha = 0.1$ the fraction of oxidation products condensed into the particle phase was $0.64 \pm 0.15$.

The losses of sulfuric acid were also calculated in the same way. Diffusion coefficient of $1*10^{-5}$ (Hanson and Eisele, 2000), $\alpha$ of 0.65 (Pöschl et al., 1998) and molar mass of 98.079 g mol$^{-1}$ were used for sulfuric acid. The fraction of sulfuric acid that

condensed into the particle phase was $0.98 \pm 0.01$. According to Lambe et al. (2011), $SO_2$ losses in the PAM are negligible. It can be concluded that the effect of precursor losses and artifacts in the PAM was not substantial in our measurements. The measurement of ammonium nitrate and ammonium sulfate is difficult because ammonia sticks on the walls of sampling systems and instruments (Suarez-Bertoa et al., 2015; Heeb et al., 2012, 2008), which may result in wall losses or an artifact on subsequent measurements. The penetration of ammonia could not be calculated, but the measured ammonium concentrations varied clearly from one case to another, implicating that the source of the ammonia was indeed the exhaust line instead of e.g., the walls of the PAM. However, longer times for the stabilization of the SP-AMS concentration would have been advantageous for the reliability of ammonium, and as a consequence, nitrate particle formation.

Karjalainen et al. (2016) and Timonen et al. (2017) estimated the effect of particle losses in a similar PAM chamber to be small. The particle losses measured by Karjalainen et al. (2016) depend on particle size and are below 10 % at the particle sizes with most particle mass. An exact calculation of the particle losses in the PAM chamber is not possible because the particle size and number increase while the aerosol sample flows through the chamber. An estimation for the particle mass losses in the chamber can be given, calculated using the average of HRLPI particle number size distributions before and after the chamber (similarly to the precursor-loss calculation by Palm et al., 2016) and the PAM particle loss curve. The particle mass loss according to this examination was $20.3 \pm 3.7$ %. Most probably, however, the actual particle mass losses in the chamber were smaller because majority of the mass actually was located at larger particle sizes that HRLPI is unable to measure, where particle losses are smaller.

No loss corrections were done based on these calculations on the results presented in this article. If corrections had been made, the presented secondary aerosol productions and production factors would be slightly higher (less than 10 %) in Figures 2 and 4-6, and in Tables 1 and 2.

## 4 Conclusions

Natural gas engines emit very little particle mass, which can make them less harmful to human health than corresponding gasoline-, diesel- or marine-fuel-oil-fueled engines. However, secondary aerosol formation also increases human exposure to aerosol particles. When natural gas engines become more common in traffic and energy production, their potential for secondary particle formation will become more important and an even more relevant object for research. Therefore, it is important to study the potential reduction of the total aerosol particle mass and related health and climate effects when shifting from liquid fuels to natural gas or biogas in combustion engines is important.

In this study, a retro-fitted natural gas engine equipped with exhaust after-treatment was studied in a laboratory in an engine test bench, using steady-state engine operation modes, i.e., constant engine speed and torque. The secondary aerosol formation was studied using a potential aerosol mass (PAM) chamber. Estimates for the atmospheric ages achieved by the PAM chamber were 4.6-10.7 days. In this study, the secondary aerosol mass potential of natural gas emission was measured to be at a small or medium level but well measurable. Compared to the primary particle mass emissions from the same engine, the secondary aerosol formation potential was substantial — approximately one to two orders of magnitude higher than the primary

aerosol mass. However, the very small primary particle masses in some of the observed engine and catalyst operation modes complicated this comparison. To give a rough estimate to the quantity of the NG engine exhaust's SOA formation potential, it was on the same level as or lower than the SOA formation potential of a diesel vehicle equipped with an oxidation catalyst or that of warm (hot start) gasoline vehicles. However, the photochemical age that was produced by the PAM chamber in our study was longer (several days) than the photochemical ages achieved in the previous studies (several hours). Therefore, the SOA formation potential must not be directly compared. Also, despite the attempts to model PAM related losses and artifacts and to estimate particle losses in PAM, the measurements performed with PAM still involve uncertainties.

The total aged aerosol, i.e., the combined primary and secondary aerosol (downstream of a PAM chamber) of the NG engine, consisted of organic matter, nitrate, sulfate and ammonium, roughly in this order. It was found that aging of the exhaust generates low-volatility organics. However, the composition of the secondary aerosol was, for the most part, inorganic; the fraction of organic matter in the secondary particles varied between 37-56 %.

Exhaust after-treatment was found to have an effect on the secondary aerosol composition. High catalyst temperature promoted the formation of sulfate particles in total aged aerosol, whereas low catalyst temperatures promoted nitrate formation. Because the amount of $NO_x$ emissions was reduced at the lower catalyst temperatures, it was concluded that the formation of nitrate in particles (total aged) depended on the ammonia concentration and sulfate particle formation rather than the $NO_x$ emissions. Sulfate and nitrate are likely to exist in the forms of ammonium sulfate and ammonium nitrate. Therefore, what limits the nitrate mass in particles is most likely the availability of ammonia which is more related to the exhaust after-treatment than fuel or combustion process.

The total aged nanoparticles formed from the natural gas exhaust were found to be less volatile than the primary particles. This can affect their lifetime in the atmosphere and therefore their impact on the radiative balance of the atmosphere or their potential to act as cloud nuclei. A higher catalyst temperature impacts the total aged particles by decreasing their volatility or by decreasing their volatile fraction.

In our study, only one constant PAM UV-light voltage could be used. With improved instrumentation, a broader variation in light intensity could be achieved, thus improving our knowledge regarding the evolution of the secondary aerosol. Also, because natural gas is not the only widely used gaseous fuel, the secondary aerosol formation potential of a more extensive fuel selection would be interesting to study. The role of lubricating oil is not known yet either – studies performed at different natural gas combustion sites and with various lubricating oils would reveal its significance to secondary aerosol formation.

## 5   Data and code availability

The data and code of this study are available from the authors upon request.

*Competing interests.*

The authors declare that they have no conflict of interest.

*Acknowledgements.* This study was funded by Tekes, the Finnish Funding Agency for Innovation, Neste, AGCO Power, Wärtsilä, Dinex Ecocat, Dekati, Suomi Analytics and Viking Line. Jenni Alanen acknowledges Gasum's gas funding for financial support. Pauli Simonen acknowledges the TUT Graduate School for its funding. Oskari Kangasniemi acknowledges the Nessling Foundation for its funding. Topi Rönkkö acknowledges the financial support from the Academy of Finland (ELTRAN project, grant number 293437).

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

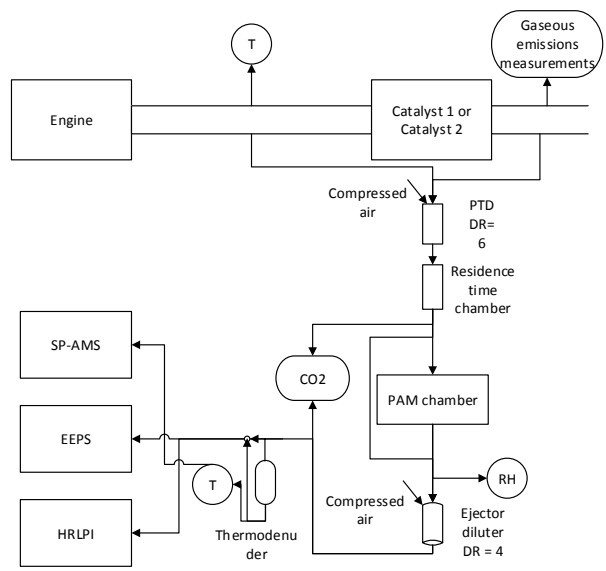

**Figure 1.** A schematic picture of the measurement setup.

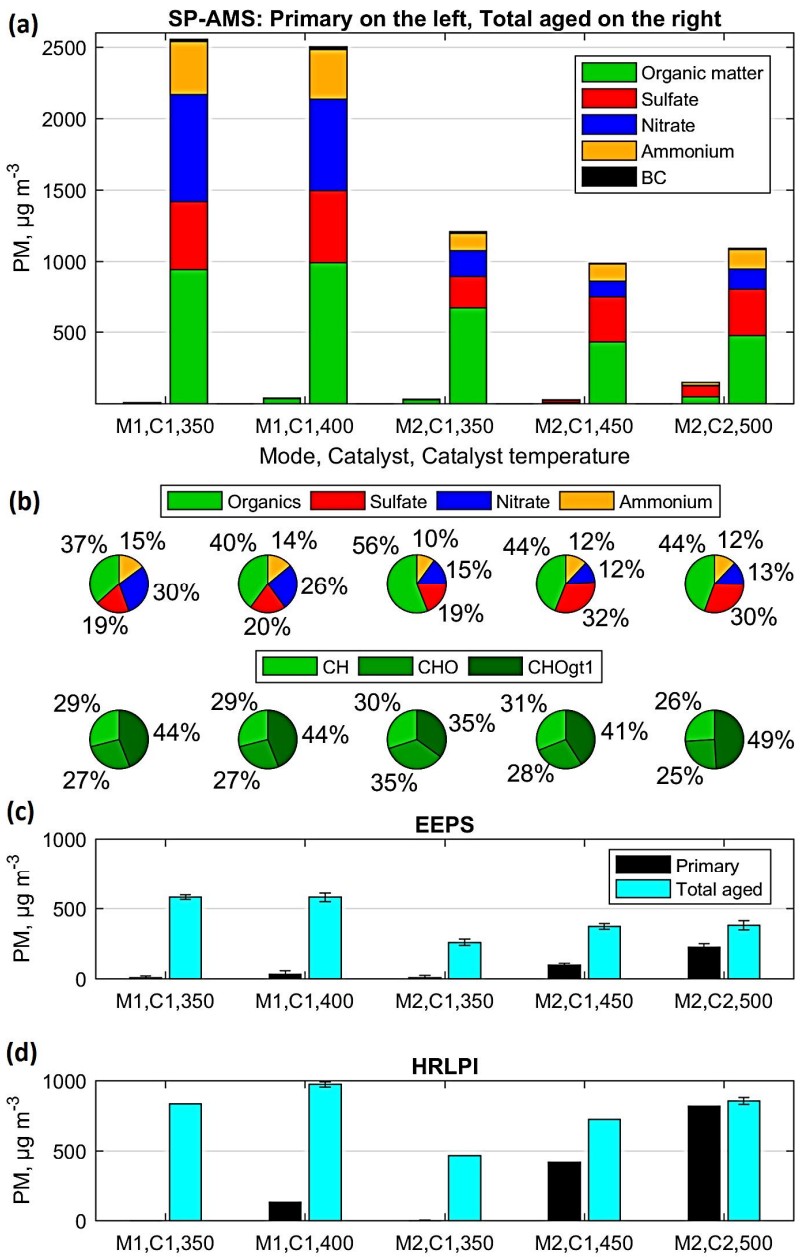

**Figure 2.** Exhaust primary and total aged particle mass concentrations measured by (a) SP-AMS, (c) EEPS and (d) HRLPI at different engine modes and catalyst temperatures. All values have been corrected by dilution ratio used in the sampling system. Secondary particle mass can be calculated by subtracting primary from total aged emission. The composition of the total aged particulate matter and the organic particulate matter is presented as pie charts (b). The fraction of black carbon is less or equal to 1 % and therefore left out from the pie charts.

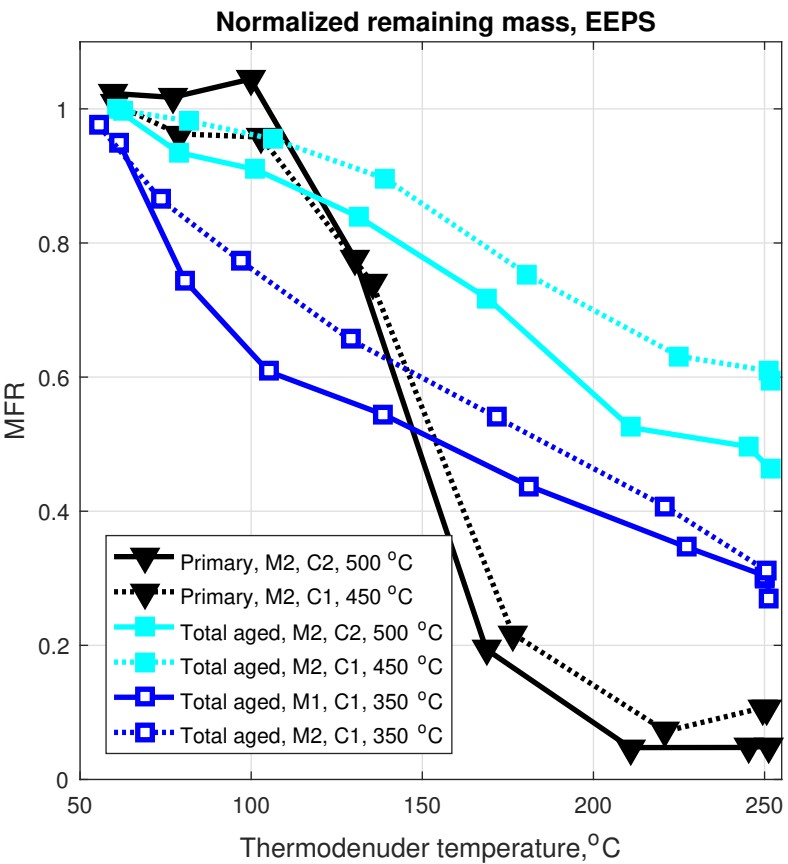

**Figure 3.** Results of particle volatility measurements. Particle mass fraction remaining (MFR) after the thermodenuder treatment for the exhaust aerosol sample of three different types of particle emission from the natural gas engine. MFR values were calculated from the size distributions measured by EEPS with unit mass assumption.

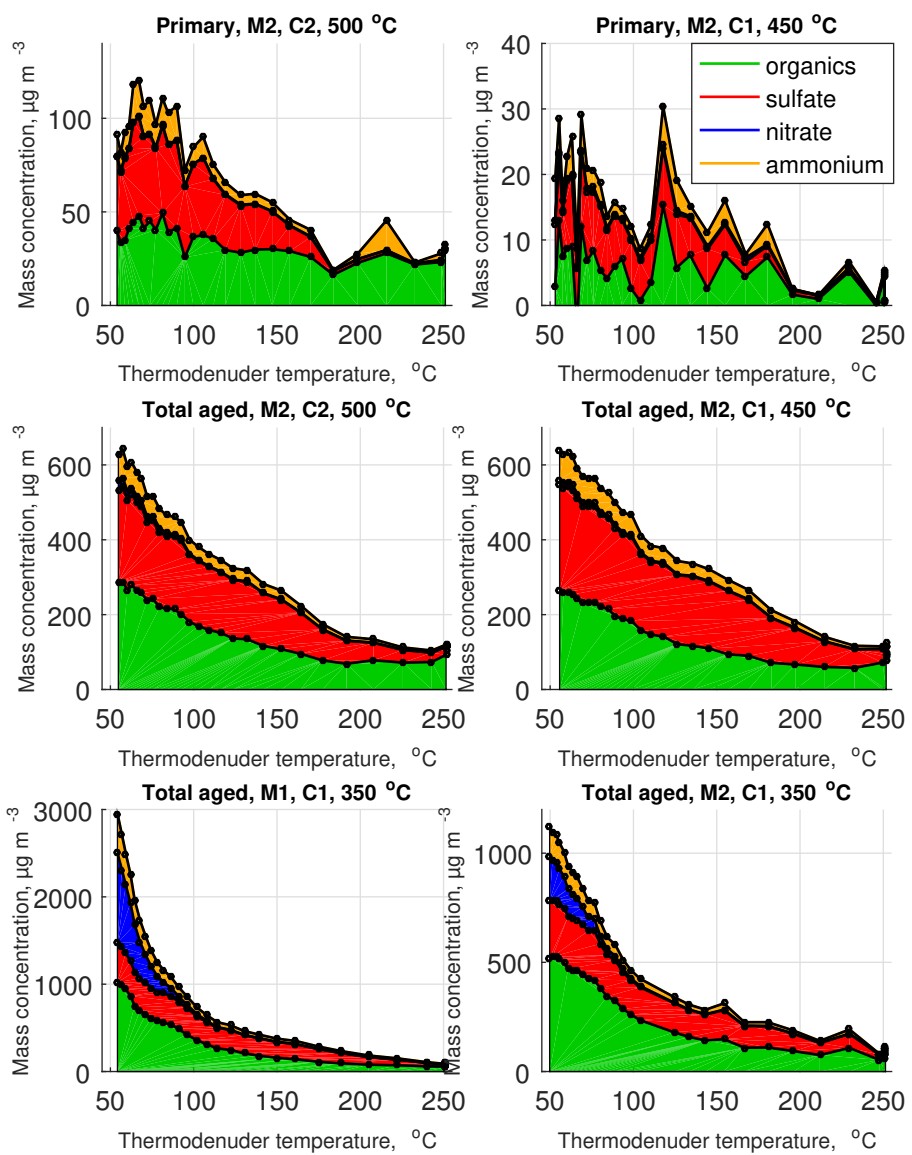

**Figure 4.** Concentration of different chemical compounds of particles remaining after the thermodenuder treatment conducted for the exhaust aerosol. The mass concentrations were measured using the SP-AMS at different thermodenuder temperatures and corrected by the dilution ratio used in the sampling system.

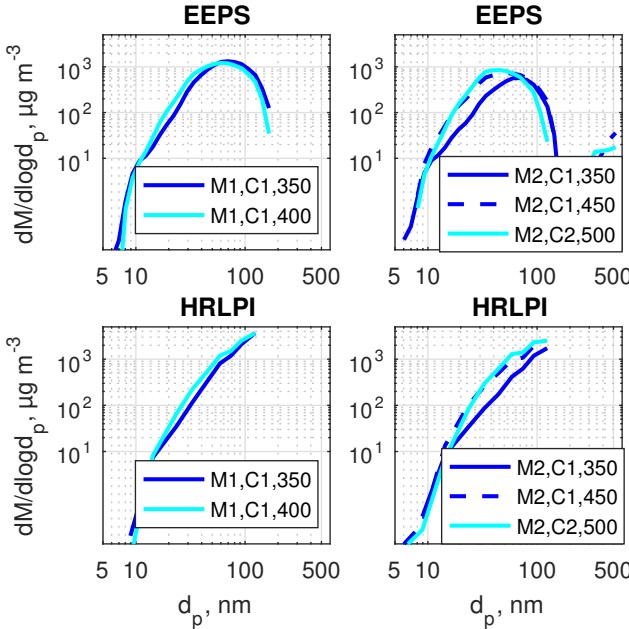

**Figure 5.** Particle mass size distributions measured by EEPS and HRLPI and corrected by the dilution ratios. Cases M1, C1 on the left, and cases M2, C1 and M2, C2 on the right. Cyan curves stand for the higher catalyst temperatures and blue for the lower ones.

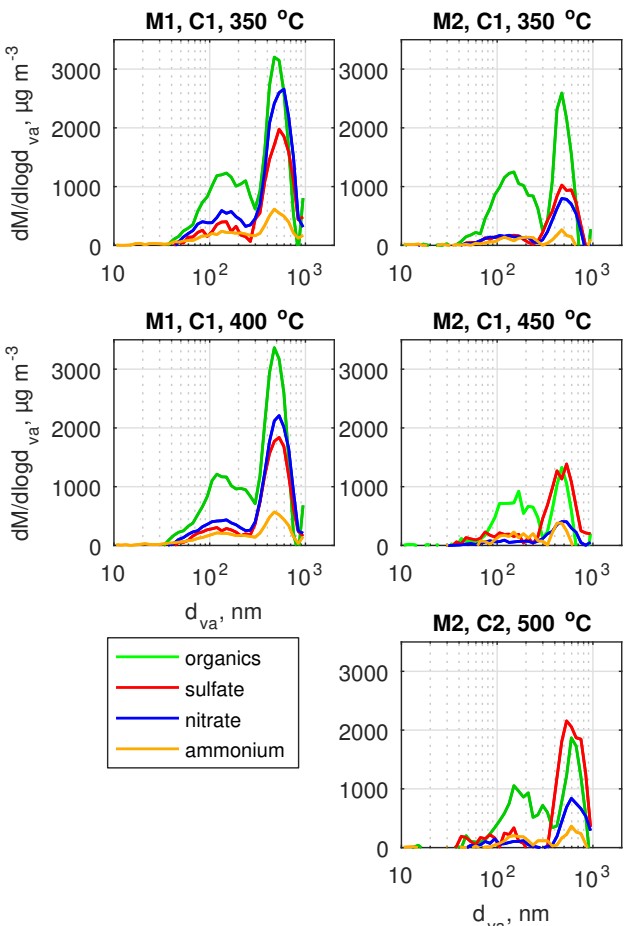

**Figure 6.** Component-wise particle mass size distributions measured by SP-AMS and corrected by the dilution ratios.

**Table 1.** (a): Particle mass concentrations of primary and total aged particles (SP-AMS), increase of particle mass (%) in PAM chamber, calculated atmospheric ages simulated by PAM chamber and O/C ratios measured by SP-AMS. If no increase in PAM is presented, it is larger than 100 000 %. (b): The particle mass of species (SP-AMS) in all cases and are also presented in the table as well as (c): the concentrations of gaseous emissions in raw exhaust (published already in Lehtoranta et al. 2016). Values have been corrected by the dilution ratio used in the sampling system.

| (a) | M1, C1, 350 °C | M1, C1, 400 °C | M2, C1, 350 °C | M2, C1, 450 °C | M2, C2, 500 °C |
|---|---|---|---|---|---|
| Primary PM, $\mu$g m$^{-3}$, SP-AMS | 9 | 40 | 31 | 28 | 150 |
| Total aged PM, $\mu$ g m$^{-3}$, SP-AMS | 2554 | 2503 | 1210 | 989 | 1093 |
| Increase in PAM, %, SP-AMS | 26800 | 6210 | 3840 | 3440 | 630 |
| Increase in PAM, %, EEPS | 7130 | 1660 | 2680 | 278 | 69 |
| Increase in PAM, %, HRLPI | - | 643 | 22800 | 75 | 4 |
| Atmospheric age, days | 10.0 | 10.7 | 4.6 | 4.7 | 9.3 |
| O/C | 1 | 1.1 | 1 | 0.9 | 1.2 |
| **(b): Total aged PM of species, $\mu$ g m$^{-3}$, SP-AMS** | | | | | |
| Organic | 944 | 993 | 669 | 430 | 476 |
| Sulfate | 475 | 502 | 228 | 317 | 330 |
| Nitrate | 749 | 641 | 182 | 115 | 143 |
| Ammonium | 372 | 348 | 119 | 121 | 135 |
| **(c): Concentrations of gaseous emissions, ppm** | | | | | |
| NO$_x$ | 3 | 4 | 3 | 12 | 4 |
| CO | 14 | 7 | 14 | 8 | 4 |
| Methane | 906 ± 16 | 904 ± 30 | 2232 ± 74 | 2238 ± 51 | 1360 ± 7 |
| Ethane | 18 | 17 | 68 | 49 | 15 |
| Propane | 1 | 1 | 21 | 6 | 1 |
| Ethene | 0 | 0 | 2 | 0 | 0 |

**Table 2.** SOA production factors calculated from the SP-AMS data in this study, and in literature (Age = OH-exposure/ $(1.5 * 10^6$ molecules $cm^{-3}$) ). Primary organic aerosol has been subtracted from the total aged organic aerosol. *Calculated assuming gasoline density 0.75 kg $l^{-1}$ and consumption $7.9\,l\,(100\,km)^{-1}$.

| Source | Age | PF (mg $kg_{fuel}^{-1}$) | Reference |
|---|---|---|---|
| NG engine: | | | |
| M1, C1, 350 °C | 10 days | 19 | This study |
| M1, C1, 400 °C | 10.7 days | 20 | This study |
| M2, C1, 350 °C | 4.6 days | 12 | This study |
| M2, C1, 450 °C | 4.7 days | 9 | This study |
| M2, C2, 500 °C | 9.3 days | 9 | This study |
| Diesel/biodiesel nonroad engine, idling | 1.5 days | 5300-12000 | Jathar et al. (2017) |
| Diesel/biodiesel nonroad engine, 50 % load | 0.8 days | 400-900 | Jathar et al. (2017) |
| Ethanol vehicle, NEDC cycle | ~1-8 days | <2 [*] | Timonen et al. (2017) |
| Gasoline vehicle, parts of NEDC cycle | ~1-8 days | 7-155 [*] | Karjalainen et al. (2016) |
| Vehicle fleet in highway tunnel | 5.4 days | 350 | Tkacik et al. (2014) |
| Gasoline vehicle, hot start | 3 h | 13.8 | Gordon et al. (2014a) |
| Gasoline vehicle, cold start | 3 h | 19-60 | Gordon et al. (2014a) |
| Gasoline vehicles, idling | 3-6 h | 5-90 | Nordin et al. (2013) |
| Gasoline vehicles, cold start | Unknown | 480 | Nordin et al. (2013) |
| Gasoline vehicle, NEDC cycle | 8 h | 345 | Platt et al. (2013) |
| Small 2-stroke off-road engine | 1-7 h | 240-1400 | Gordon et al. (2013) |
| Small 4-stroke off-road engine | 1 h | 100-130 | Gordon et al. (2013) |
| Diesel vehicle, deactivated catalyst | Unknown | 230-560 | Chirico et al. (2010) |
| Diesel vehicle, catalyst working | Unknown | 12-20 | Chirico et al. (2010) |

**Table 3.** The temperatures where 50 % of the volatile fraction of species has evaporated.

| $T_{volatile, 50\%}$ (°C) | Primary M2, C2,500 °C | Primary M2, C1, 450 °C | Total aged M2, C2, 500 °C | Total aged M2, C1, 450 °C | Total aged M1, C1, 350 °C | Total aged M2, C1, 350 °C |
|---|---|---|---|---|---|---|
| Organics | 115 | - | 99 | 104 | 87 | 93 |
| Sulfate | 125 | - | 152 | 168 | 120 | 147 |
| Nitrate | 72 | - | 80 | 84 | 65 | 65 |
| Ammonium | 104 | - | 97 | 112 | 70 | 79 |