# Peer review of "Comparison of primary and secondary particle formation from natural gas engine exhaust and of their volatility characteristics"

_Atmospheric Chemistry and Physics, 2017_

## Referee Comment (RC1) · Anonymous Referee #1 · 30 Mar 2017

The authors simulated the effects of atmospheric aging of exhaust emissions from a natural gas engine with a PAM chamber and investigated the volatility of fresh and aged particles using a thermodenuder. The authors show that the engine tested produced significant secondary particulate matter with respect to the primary emission and that composition and volatility of the secondary aerosol was influenced by engine/ catalyst temperatures. The authors present a clear motivation for their work and the article is generally well written. However, there are major issues, which should preclude publication at this stage, outlined as follows:

Major comments

Scope of the paper: The authors are using a relatively new, hence unestablished,

technique to investigate emissions from one engine during conditions not shown to be relevant to real world driving. They then use their results to suggest that "the shift from traditional liquid fuels to natural gas can have a decreasing effect on total particle pollution in the atmosphere". They also do not quantify the 'shift' to which they refer. Given these obvious limitations, I do not feel that such strong statements on the effect of natural gas engines on the atmosphere are justified. I would suggest they remove them from the manuscript.

Wall effects and sampling artefacts: There are issues relating to the measurement of ammonia and ammonium nitrate. In the PAM, ammonium nitrate will form in the presence of NOX and ammonia. If sulfate is present, ammonium sulfate will form preferentially before ammonium nitrate. However ammonia is a very 'sticky' compound and difficult to measure see e.g. Suarez-Bertoa et al. It is thus clear that losses of gaseous ammonia and the walls of the sampling system and reactor will have a profound impact on the observed secondary inorganics, and potentially on subsequent experiments as the ammonia slowly desorbs. In turn, the effects observed in this paper may very well relate more to sampling conditions e.g. flow rates/ temperatures than to processes occurring in the atmosphere or in the engine. Ideally, the authors should determine how long it takes for ammonia concentrations reaching the PAM to match those emitted at the tailpipe and how long it takes for the PAM to become ammonia free after an experiment. Meanwhile, ammonium nitrate has been shown to influence measurement of the $CO_2+$ fragment in the AMS by Pieber et al. This influence on $CO_2+$ is small, 0.4-10% of the nitrate mass, but is instrument dependent, and becomes larger when the error is propagated through the AMS fragmentation table to mz 28 and so on. The O/C ratio is also affected. This may go part way to explaining the observed difference between the EEPS and AMS. Fortunately, this may be corrected for retrospectively using the $CO_2+$ signal from the initial ammonium nitrate calibration, and is recommended for measurements such as these, with a high ammonium nitrate fraction. How do wall losses in the PAM effect the observed organic mass concentrations? This is of course not easy to answer, but some consideration of the effect of particle losses and vapour

losses (organic/ inorganic secondary precursors) is required.

Minor Comments

Pg 1 Ln 8: What does low or moderate mean in this context? It seems to contradict Pg 1 Ln 10: "particle mass measured downstream the PAM chamber, was 6-184 times as high as the mass of the emitted primary exhaust particles." Presumably, the authors mean in absolute mass, but please clarify.

Pg 1 Ln 20: This last statement is too strong given the limited scope of the study (See major comments).

Pg 2, introduction: The recent review of SOA from vehicles by Gentner et al. should be cited somewhere here.

Pg 3 Ln 31: Jathar et al. found that SOA formation from raw LNG is higher than from gasoline. This result is relevant to this introduction and to the discussion sections of this work.

Pg 4 Ln 9: What was the remaining 3% of the fuel?

Pg 4 ln 11: How relevant are these engine conditions to real world driving?

Pg 4 Ln 26: Was the sampling system heated in any way? If not, how did this effect losses of secondary precursors?

Pg 6 Ln 12: How was $CO_2$ gas interference at mz 44 corrected for?

Pg 6 Ln 34: 'Representative' of what? What do the other cases look like? A figure similar to Figure 4 which shows 'non-representative' results should be shown in the SI.

Pg 11 Ln 5: "With a decreasing catalyst temperature, the mass concentration and fraction of sulfate in total aged particles decreased (Fig. 4). This was expected; at lower catalyst temperatures, the oxidation of $SO_2$ to $SO_3$ decreases and less sulfuric acid (sulfates) can form (Arnold et al., 2012). The mass fraction of nitrate in secondary particles increased with a decreasing catalyst temperature. This could not be explained by catalyst performance improvement: gaseous NOx levels remained similar in all catalyst temperatures or rose with an increasing catalyst temperature (see Lehtoranta et al., 2016). Sulfate concentrations, however, could explain the behaviour of nitrate concentrations: If enough gaseous sulfuric acid is available, ammonium sulfate forms, and if not, more ammonium nitrate can form instead." How large was the decrease in sulfate (absolute mass) shown in Figure 4? According to table 1 this decrease is small for the SP-AMS. In quantative/ stoichiometric terms is the decrease in mass of sulfate sufficient to explain the increase in ammonium nitrate? If the change in sulfate mass is rather small, and since the authors are using catalysts with urea reduction, shown to result in increasing ammonia selectivity (vs. NO) at lower temperature, is it not more likely that an increase in ammonia emissions at low temperatures causes the observed effects?

Pg 11 Ln 22. Why was a collection efficiency of 0.5 used? This is clearly suspect given the discrepancy between EEPS and AMS. What, in quantative terms ,would be the time dependent collection efficiency using the parametrization by Middlebrook and how would this effect the magnitude of the SOA formation and SOA/POA ratios?

Pg 12 Ln 1. The authors extrapolate from one natural gas engine to natural engines in general (see major comments).

References

1 Suarez-Bertoa, R. et al. Intercomparison of real-time tailpipe ammonia measurements from vehicles tested over the new world-harmonized light-duty vehicle test cycle (WLTC). Environmental Science and Pollution Research 22, 7450-7460 (2015). 2 Pieber, S. et al. Inorganic Salt Interference on CO2+ in Aerodyne AMS and ACSM Organic Aerosol Composition Studies. Environmental Science and Technology 50, 10494–10503 (2016). 3 Gentner, D. R. et al. A review of urban secondary organic aerosol formation from gasoline and diesel motor vehicle emissions. Environmental

Science & Technology (2016). 4 Jathar, S. H. et al. Secondary organic aerosol formation from photo-oxidation of unburned fuel: Experimental results and implications for aerosol formation from combustion emissions. Environmental Science and Technology 47, 12886-12893 (2013). 5 Heeb, N. V. et al. Three-way catalyst-induced formation of ammonia—velocity-and acceleration-dependent emission factors. Atmospheric Environment 40, 5986-5997 (2006). 6 Middlebrook, A. M., Bahreini, R., Jimenez, J. L. & Canagaratna, M. R. Evaluation of composition-dependent collection efficiencies for the aerodyne aerosol mass spectrometer using field data. Aerosol Science and Technology 46, 258-271 (2012).

---

## Referee Comment (RC2) · Anonymous Referee #2 · 4 Apr 2017

Alanen et al. studied the secondary particle formation potential from natural gas engine exhaust using a potential aerosol mass (or an oxidation flow) reactor and concluded that the secondary particle formation was (i) substantially higher than the primary aerosol emissions, (ii) dominated by organic compounds but had significant contributions from inorganic compounds, (iii) dependent on the catalyst type and temperature, and (iv) lower than the secondary particle formation observed for gasoline and diesel vehicles.

The manuscript provides strong motivation for the study and the instrument- and analysis-related methods employed for this study are adequate. My concern with the methods is the operation and use of the oxidation flow reactor (see major comment (1)

below). The manuscript, for the most part, is well written and the results are robust. However, I am hesitant on accepting the manuscript for publication in its current form. I would like the authors to consider the following major and minor comments:

Major comments:

1. Operation and use of the oxidation flow reactor (OFR): I have two concerns surrounding the use of the PAM or the OFR. One, the authors have not examined the formation of secondary particles as a continuous function of the photochemical age, in contrast to previous studies. This is a problem because, as has been shown previously, secondary particle formation increases with photochemical age at low ages but eventually decreases (presumably from gas-phase fragmentation reactions and heterogeneous chemistry) at higher ages. Since the current study does not simulate this continuous evolution, it is unclear if the authors would have seen a pattern similar to that observed with other OFR studies. Do the authors have that data? Can the study be repeated at least for one of the engine load-catalyst conditions? More importantly, this creates the problem of comparing the data obtained in this study with other chamber and OFR studies (that have quantified secondary particle formation on at least a semi-continuous basis for photochemical age) and presents challenges in making conclusions about the natural gas engine (see next major comment below). Second, the authors have not considered known artifacts of using an OFR related to the loss of vapors to the OFR walls and accelerated chemistry (Palm et al., 2016) and the short residence times and small condensation sinks (Jathar et al., 2017) in an OFR that might not allow the oxidation products to condense as secondary particles (since the natural gas engine produced very few primary particles, I think this might be an important factor). All of these contribute to underestimating the secondary particle formation in OFRs, again affecting the comparison with chamber studies. A discussion of these effects and an attempt to correct for these effects will tell the authors if and how these artifacts could affect the results and conclusions. I would refer the authors to Palm et al. (2016) for methods to correct for these artifacts. In addition, a brief discussion of the

pitfalls in using the OFR (e.g., accelerated chemistry leading to multiple reaction steps in the gas-phase) and particularly its comparison to chamber data would be helpful.

2. Comparison with earlier studies: Based on the above comment, I have a few questions that center around the conclusions of this study: (i) can the results of this study at high photochemical ages (several days) be compared to chamber studies of gasoline and diesel vehicles performed over several hours of photochemistry (in my opinion, probably not)? (ii) how would the OFR artifact corrections change the comparison described in Table 2 (the Table could also add recent measurements made by Jathar et al., 2017 for diesel engines)? (iii) how confident are the authors in claiming that the secondary particle production from natural gas engines is quite small compared to gasoline and diesel engines, especially in urban areas where photochemical ages are low (0.5-1.5 days). (ii) how relevant are the high photochemical age results in this study to the atmosphere given that the there are other processes (e.g., transport, deposition) that are relevant at the same time scale? The operation of the OFR and the artifacts linked to the OFR do not allow for definitive answers to any of the above questions. This fact needs to be considered for the summary/conclusions from this work.

3. Style and phrasing: The writing, while detailed, needs to be paid more attention and the style and phrasing need to be improved throughout the manuscript. Here are just a few examples: (i) Page 11, line 30: 'Although' instead of 'However, although', (ii) Page 11, line 8: 'remained similar at all catalyst temperatures' instead of 'remained similar in all catalyst temperatures', (iii) Page 10, line 29: 'waited' instead of 'was waited', (iv) Page 7, line 6: 'condense' instead of 'condensate'.

Minor comments:

1. Page 2, line 27: Is there a reference for the comment on diesel vehicles? Also, can this comment be made for other combustion/energy sources?

2. Page 2, line 30: May be use the word 'oxidative catalyst' to refer to the aftertreatment device?

3. Page 3, line 4: Is there a newer reference instead of Weitkamp for the last sentence? Examples: Gordon et al., 2014 or Jathar et al., 2017.

4. Page 4, lines 9-14: It would help if there was a brief mention of the size, horsepower, typical application, on- or off-road characteristics for the natural gas engine used.

5. Page 4, lines 15-24: What is the rationale for using two different aftertreatment configurations and the different temperatures? Is it based on the drive cycle expected for these engines? It is likely this fact is hidden in some of the earlier literature by the group but it would be beneficial to mention it here.

6. Page 5, lines 9-29: This is a novel application of the MCM to determine photochemical exposure given that exhaust reactivity can significantly suppress OH exposure. Peng et al. (2015) and Li et al. (2015) have developed empirical relationships to calculate OH exposure based on the mixture reactivity, residence time, and humidity in OFRs. Can the authors comment on why the method of Peng+Li was not used to calculate OH exposure? Or in other words, is there reason to believe that the MCM offers better predictions? Has the MCM output been evaluated against OH exposure measurements for an OFR application?

7. Page 5, lines 30-34: I am not sure I understand how the RH in the PAM chamber was as high as 80% when the clean dilution air (which was dry or 30%) accounted for 24 parts by volume to the exhaust. The mass balance on the water vapor does not seem to work out. The only way to explain the result is that water vapor is being generated in the PAM?

8. Page 6, lines 27-31: Subtraction of primary particle mass needs to be done when presenting the secondary particle production factors. For instance, SOA emission factors in Table 2. Do the authors know of the recent work of Link et al. (2017) where they examined photochemical production of inorganic compounds from light-duty vehicle exhaust? That might be another study to incorporate within the inorganic particle production discussion.

9. Page 7, lines 17-21: Long sentences beginning with 'It is possible' and ending with 'into the atmosphere'. Consider breaking into multiple sentences.

10. Page 8, lines 14-18: Why isn't the standard calculation of EF based on the $CO_2$ concentrations and carbon intensity of the fuel, employed in this work?

11. Page 9, line 21: While I understand what the authors mean, 'oxidated alpha-pinene particles' is not correct.

12. Figure 2: The pie charts are too small. Please consider increasing them in size. Will the plots a, c, and d benefit from being in emission factor units?

13. Figures 5 and 6: Do the EEPS and HRLPI size distributions compare well with the AMS size distributions? Can the EEPS, HRLPI, and the AMS data be combined to provide a more complete picture of the size- and composition-resolved emission/production of particles?

14. Can the authors speculate on the species in the exhaust and photochemical mechanisms that can explain the secondary organic particle production?

---

## Author Comment (AC1) · 12 May 2017

**Relevant changes to manuscript acp-2017-44:**

**Dear reviewers and editor,**

We thank the reviewers for taking time to read our paper and for their insightful and constructive comments.

Based on the reviewer comments we have added a section that focuses on PAM losses and artifacts, we have highlighted the role of ammonia in interpretation of our results and reduced the direct comparison of our results with the existing literature. We also re-handled the data using a new evaluation of SP-AMS collection efficiency and we added one figure to the supplementary section, discussing the mass size distributions of the three aerosol instruments and their differences. Furthermore, the manuscript was proof-read by a native English speaker and multiple corrections were made to the style and phrasing. Also, e.g., several new citations were made.

In this document, there are point-to-point answers to the reviewers. We have copied the reviewer comments with black text. Our responses are in regular blue font and *the changes made in the manuscript are marked with italic blue text.*

Attached, there is the revised version of our manuscript, the revised supplementary section and file where the made changes are marked with red cross-out or blue underlining.

On behalf of the authors,

Sincerely,

Jenni Alanen

**Anonymous Referee #1**

The authors simulated the effects of atmospheric aging of exhaust emissions from a natural gas engine with a PAM chamber and investigated the volatility of fresh and aged particles using a thermodenuder. The authors show that the engine tested produced significant secondary particulate matter with respect to the primary emission and that composition and volatility of the secondary aerosol was influenced by engine/ catalyst temperatures. The authors present a clear motivation for their work and the article is generally well written. However, there are major issues, which should preclude publication at this stage, outlined as follows:

Major comments

Scope of the paper: The authors are using a relatively new, hence unestablished, technique to investigate emissions from one engine during conditions not shown to be relevant to real world driving. They then use their results to suggest that "the shift from traditional liquid fuels to natural gas can have a decreasing effect on total particle pollution in the atmosphere". They also do not quantify the 'shift' to which they refer. Given these obvious limitations, I do not feel that such strong statements on the effect of natural gas engines on the atmosphere are justified. I would suggest they remove them from the manuscript.

The Referee is right, the statement was too strong and the shift was not validated. The sentence has been removed. Also other statements on the effect of natural gas engines on the atmosphere have been modified or removed.

Wall effects and sampling artefacts: There are issues relating to the measurement of ammonia and ammonium nitrate. In the PAM, ammonium nitrate will form in the presence of NOX and ammonia. If sulfate is present, ammonium sulfate will form preferentially before ammonium nitrate. However ammonia is a very 'sticky' compound and difficult to measure see e.g. Suarez-Bertoa et al. It is thus clear that losses of gaseous ammonia and the walls of the sampling system and reactor will have a profound impact on the observed secondary inorganics, and potentially on subsequent experiments as the ammonia slowly desorbs. In turn, the effects observed in this paper may very well relate more to sampling conditions e.g. flow rates/ temperatures than to processes occurring in the atmosphere or in the engine. Ideally, the authors should determine how long it takes for ammonia concentrations reaching the PAM to match those emitted at the tailpipe and how long it takes for the PAM to become ammonia free after an experiment.

We don't believe that the effects on ammonium and nitrate only relate to sampling conditions but to catalyst (or exhaust) temperature and engine operation resulting higher ammonia concentrations in the exhaust. We admit, however, that ammonia is difficult to measure, a fact which should be pointed out in the manuscript.

It was not possible to determine the time that it took the ammonia to saturate in the sampling system or PAM or the time it took to become ammonia free in a way the Referee desires. Ammonia measurements in the raw exhaust showed below detection limit of 2 ppm for all the stable engine operation modes that were tested. When clean compressed air was lead though the PAM with UV lights on, ammonium level – measured by SP-AMS – dropped to zero concentration in less than ten minutes. This, however, is not sufficient test to determine how long it takes for the PAM to become ammonia free after an experiment because of the lower relative humidity of the compressed air and because ammonia could exist in gaseous phase (we did not measure ammonia in diluted exhaust) and in particles below SP-AMS detection limit even if the SP-AMS did not detect it.

Following changes were made to the manuscript:

Page 12: *We note that because nitrate formation is limited by ammonium, the slow stabilization is probably related to ammonia.*

Page 14: *The measurement of ammonium nitrate and ammonium sulfate is difficult because ammonia sticks on the walls of sampling systems and instruments (Suarez-Bertoa et al., 2015; Heeb et al., 2012, 2008), which may result in wall losses or an artifact on subsequent measurements. The penetration of ammonia could not be calculated, but the measured ammonium concentrations varied clearly from one case to another, implicating that the source of the ammonia was indeed the exhaust line instead*

*of e.g., the walls of the PAM. However, longer times for stabilization of the SP-AMS concentration would have been advantageous for the reliability of ammonium, and as a consequence, nitrate particle formation.*

-- Meanwhile, ammonium nitrate has been shown to influence measurement of the CO2+ fragment in the AMS by Pieber et al. This influence on CO2+ is small, 0.4-10% of the nitrate mass, but is instrument dependent, and becomes larger when the error is propagated through the AMS fragmentation table to mz 28 and so on. The O/C ratio is also affected. This may go part way to explaining the observed difference between the EEPS and AMS. Fortunately, this may be corrected for retrospectively using the CO2+ signal from the initial ammonium nitrate calibration, and is recommended for measurements such as these, with a high ammonium nitrate fraction.

We agree with the Referee that the influence of inorganic species on CO2+ signal is very important topic regarding AMS data analysis, especially in the case of laboratory engine emissions, and it was not mentioned in the manuscript. We investigated the impact of ammonium nitrate on CO2+ signal with the calibration performed in the laboratory after the measurement campaign. According to the calibration, nitrate to CO2+ resulted in a slope (b) of 0.016. Compared to the paper of Pieber et al. (2016) b in our instrument was in the lower end of the range of b values for 8 tested instruments (0.004-0.102, P10-P90). Roughly estimating from Figure 5 in Pieber et al. (2016) the impact of ammonium nitrate interference on O/C and H/C ratios was less than 5% in our data (with NO3/OA <0.85; O/C >0.7). Therefore, we decided that we don't apply a correction for Org, O/C or H/C data because of ammonium nitrate interference.

Following changes were made to the manuscript:

Page 6: *The impact of ammonium nitrate interference on CO2+, O/C and H/C ratios was evaluated to be small, less than 5 % (Pieber et al., 2016). Therefore, a correction of ammonium nitrate interference was not applied for organics or O/C data.*

-- How do wall losses in the PAM effect the observed organic mass concentrations? This is of course not easy to answer, but some consideration of the effect of particle losses and vapour losses (organic/ inorganic secondary precursors) is required.

Based on the model of Palm et al. (2016), which deals with the fate of condensable gases in an oxidation flow chamber, wall losses in the PAM were not substantial for condensable organic oxidation products or sulfuric acid. According to the model, the accelerated chemistry in the PAM was a more potential source of losses, but also those losses were small. Particle losses, measured by Karjalainen et al. (2016), were also considered. An exact calculation of particle losses is not possible because of new particle formation and particle growth in the PAM and the dependence of particle losses on particle size, but an estimation for particle losses was calculated.

A new chapter concerning particle and precursor losses was added in pages 13-14:

*The losses of condensable organic oxidation products and artifact effects of the accelerated chemistry in the PAM have been evaluated following the method of Palm et al. (2016) for the cases in Figure 2. HRLPI number size distributions were used to calculate the condensation sink needed in the loss calculation. Molar mass of 200 g mol$^{-1}$, diffusion coefficient of $7*10^{-6}$ m$^2$ s$^{-1}$ (Tang et al., 2015) and rate constant for reaction with OH of $1*10^{-11}$ (Ziemann and Atkinson, 2012) were applied. For sticking coefficient selection α = 1 (assumed by Palm et al. 2016), the fraction of oxidation products condensed on particle phase was 0.94 ± 0.03, but for α = 0.1 the fraction of oxidation products condensed on particle phase was 0.64 ± 0.15.*

*The losses of sulfuric acid were also calculated in the same way. Diffusion coefficient of $1*10^{-5}$ (Hanson and Eisele, 2000), α of 0.65 (Pöschl et al., 1998) and molar mass of 98.079 g mol$^{-1}$ were used for sulfuric acid. The fraction of sulfuric acid that condensed on particle phase was 0.98 ± 0.01. According to Lambe et al. (2011), SO2 losses in the PAM are negligible. It can be concluded that the effect of precursor losses and artifacts in the PAM was not substantial in our measurements. The measurement of ammonium nitrate and ammonium sulfate is difficult because ammonia sticks on the walls of sampling systems and instruments (Suarez-Bertoa et al., 2015; Heeb et al., 2012, 2008), which may result in wall losses or an artifact on subsequent measurements. The penetration of ammonia could not be calculated, but the measured ammonium concentrations varied clearly from one case to another, implicating that the source of the ammonia was indeed the exhaust line instead of e.g., the walls of the PAM. However, longer times for stabilization of the SP-AMS concentration would have been advantageous for the reliability of ammonium, and as a consequence, nitrate particle formation.*

*Karjalainen et al. (2016) and Timonen et al. (2017) estimated the effect of particle losses in a similar PAM chamber to be small. The particle losses measured by Karjalainen et al. (2016) depend on particle size and are below 10 % at the particle sizes with most particle mass. An exact calculation of the particle losses in the PAM chamber is not possible because the particle size and number increase while the aerosol sample flows through the chamber. An estimation for the particle mass losses in the chamber can be given, calculated using the average of HRLPI particle number size distributions before and after the chamber (similarly to the precursor-loss calculation by Palm et al., 2016) and the PAM particle loss curve. The particle mass loss according to this examination was 20.3 ± 3.7 %. Most probably, however, the actual particle mass losses in the chamber were smaller because majority of the mass actually was located at larger particle sizes that HRLPI is unable to measure, where particle losses are smaller.*

A comment on particle losses and their effect on the differences of the results of the instruments were also added:

Page 12: *Also, particle losses may play a role in the differences between instruments; particle losses in the PAM chamber were larger in the HRLPI size range than in the SP-AMS size range. Nevertheless, most probably, the largest role was played by the differences in instrument size ranges.*

Minor Comments

Pg 1 Ln 8: What does low or moderate mean in this context? It seems to contradict Pg 1 Ln 10: "particle mass measured downstream the PAM chamber, was 6-184 times as high as the mass of the emitted primary exhaust particles." Presumably, the authors mean in absolute mass, but please clarify.

Yes, absolute mass was meant, compared with the values in the literature. Because of the referee comments received from both Referees, the sentence was modified to not give direct evaluation of the magnitude of the secondary aerosol formation potential. The ordering of the sentences in the abstract were also changed.

Pg 1: *The PAM chamber was used with a constant UV-light voltage, which resulted in relatively long equivalent atmospheric ages of 11 days at most. The studied retrofitted natural gas engine exhaust was observed to form secondary aerosol. The mass of the total aged particles, i.e., particle mass measured downstream of the PAM chamber, was 6-268 times as high as the mass of the emitted primary exhaust particles.*

Pg 1 Ln 20: This last statement is too strong given the limited scope of the study (See major comments).

The conclusion in the last sentence was softened and totally reformulated.

Pg 1: *According to the results of this study, the exhaust of a natural gas engine equipped with a catalyst forms secondary aerosol when the atmospheric ages in a PAM chamber are several days. The secondary aerosol matter has different physical characteristics from those of primary particulate emissions.*

Pg 2, introduction: The recent review of SOA from vehicles by Gentner et al. should be cited somewhere here.

The review was read and cited as requested. Also the study of Goyal and Sitharta was cited in the introduction section.

Pg 2: *In general, vehicles emit substantial fraction of anthropogenic precursors for SOA formation (Gentner et al., 2017), and the amount of potential SOA often exceeds the emissions of primary organic aerosol.*

Pg 4: *Goyal and Sidhartha (2003) recorded a notable improvement in the air quality of Delhi when a portion of vehicles were changed to natural gas vehicles in 2001.*

Pg 3 Ln 31: Jathar et al. found that SOA formation from raw LNG is higher than from gasoline. This result is relevant to this introduction and to the discussion sections of this work.

Unfortunately, we don't fully understand this comment. Jathar et al. did not study LNG but Fischer Tropsch from NG. Fischer-Tropsch from natural gas is a liquid fuel, and as such, not directly comparable with natural gas.

Pg 4 Ln 9: What was the remaining 3% of the fuel?

The natural gas properties have been described in more detail by Alanen et al. (2015): "The fuel used was Russian pipeline natural gas with high methane content. It contained 97.2 vol% methane, 1.37% ethane, 0.17% propane, 0.07% other hydrocarbons, 0.9% nitrogen and 0.2% carbon dioxide."

Following changes were made to the manuscript:

Pg 4: *The engine used was a retrofitted spark ignition natural gas engine using Russian pipeline natural gas as fuel. Methane content of the fuel was 97 %, other hydrocarbon content was 1.6 % and nitrogen content was 0.9 %. Sulfur content was below 1 ppm.*

Pg 4 ln 11: How relevant are these engine conditions to real world driving?

The two operation modes simulated typical natural gas power plant operation with the retrofitted passenger car engine and thus did not directly represent real world driving.

The manuscript was slightly modified.

Pg 4: *A small (2.0 liter displacement) spark ignited passenger car engine was used for the measurements with Russian pipeline natural gas as fuel. The methane content of the fuel was 97 %, other hydrocarbon content was 1.6 % and nitrogen content was 0.9 %. The sulfur content was below 1 ppm. The engine was run at two steady-state engine operation modes with torque of 70 Nm and speed of 2700 rpm (Mode 1, 15 M1) and torque of 35 Nm and speed of 3100 rpm (Mode 2, M2). In engine mode 2, short chain hydrocarbons were added into the exhaust to make it resemble the exhaust of a power plant NG engine. The exhaust gas composition in two operation modes simulated typical natural gas power plant exhaust gas composition. The engine, natural gas and lubricating oil properties as well*

*as the engine operation modes have been described in more detail by Murtonen et al. (2016), Alanen et al. (2015) and Lehtoranta et al. (2016).*

Pg 4 Ln 26: Was the sampling system heated in any way? If not, how did this effect losses of secondary precursors?

The porous tube diluter (PTD) dilution air was heated to 30 °C in order to achieve constant dilution conditions. The PTD has been designed to perform without particle losses by radial entrainment of the dilution air (Ntziachristos et al. 2016; Mikkanen and Moisio 2001). The losses of secondary precursors in the PTD and the following residence time chamber have not been defined. They were similar in all the studied cases because the sampling setup was not changed.

Ntziachristos, L., Saukko, E., Lehtoranta, K., Rönkkö, T., Timonen, H., Simonen, P. and Karjalainen, P.: Particle emissions characterization from a medium-speed marine diesel engine with two fuels at different sampling conditions (2016), Fuel, 186, 456–465.

Following changes were made to the manuscript:

Pg 5: *For the particle measurement instruments, the sampling system consisted of a porous tube diluter (PTD, Mikkanen and Moisio 2001; Ntziachristos et al. 2004) with dilution ratio (DR) 6, followed by a residence time chamber with a residence time 6 s. The dilution air was heated to 30 ◦C in order to achieve constant dilution conditions. A second dilution stage was carried out with an ejector diluter (Dekati Ltd.) with DR 4.*

Pg 6 Ln 12: How was $CO_2$ gas interference at mz 44 corrected for?

AMS $CO_2$ gas interference was corrected by using the measured $CO_2$ concentrations. One sentence was added to the manuscript.

Pg 6: *The $CO_2$ gas interference in the AMS data was corrected by using the $CO_2$ concentrations measured with Sick Maihak SIDOR gas analyzer.*

Pg 6 Ln 34: 'Representative' of what? What do the other cases look like? A figure similar to Figure 4 which shows 'non-representative' results should be shown in the SI.

The word choice was wrong. The cases cover all the tested exhaust temperatures and both engine operation modes. In the cases in Figure 2 we have measured good quality data with all the instruments for both primary emission and secondary aerosol formation potential.

Following changes were made to the manuscript:

Pg 7: *The cases (engine mode, catalyst, catalyst temperature) included in this paper cover all the tested exhaust temperatures and both engine operation modes and have data collected with all available instruments at both primary and total aged aerosol measurements.*

Pg 11 Ln 5: "With a decreasing catalyst temperature, the mass concentration and fraction of sulfate in total aged particles decreased (Fig. 4). This was expected; at lower catalyst temperatures, the oxidation of $SO_2$ to $SO_3$ decreases and less sulfuric acid (sulfates) can form (Arnold et al., 2012). The mass fraction of nitrate in secondary particles increased with a decreasing catalyst temperature. This could not be explained by catalyst performance improvement: gaseous $NO_x$ levels remained similar in all catalyst temperatures or rose with an increasing catalyst temperature (see Lehtoranta et al., 2016). Sulfate concentrations, however, could explain the behaviour of nitrate concentrations: If enough gaseous sulfuric acid is available, ammonium sulfate forms, and if not, more ammonium nitrate can form instead." How large was the decrease in sulfate (absolute mass) shown in Figure 4?

According to table 1 this decrease is small for the SP-AMS. In quantative/ stoichiometric terms is the decrease in mass of sulfate sufficient to explain the increase in ammonium nitrate? If the change in sulfate mass is rather small, and since the authors are using catalysts with urea reduction, shown to result in increasing ammonia selectivity (vs. NO) at lower temperature, is it not more likely that an increase in ammonia emissions at low temperatures causes the observed effects?

Thank you for a very good comment. The interpretations in the quotations were clearly not justified. Ammonium had an effect on nitrate concentrations. In fact, nitrate concentration correlated well with ammonium ($R^2$=0.986 among all the total aged cases). However, the explanation that ammonia emissions would have increased at lower temperatures is not necessarily supported by the NOx reduction data because the NOx emissions either did not depend on the catalyst temperature or decreased at lower temperatures.

The decrease in sulfate was 5-30 %, which was not large but in the same direction in all cases. In stoichiometric terms, in the M1, C1 case the decrease in sulfate mass could explain only 32% of the increase in nitrate. In this case, however, ammonium could not explain the increase either because the ammonium concentration decreased with increasing nitrate. By contrast, in the case M2, C1 the nitrate increase could be more than fully explained by sulfate decrease.

Even though the secondary inorganic aerosol concentrations were measured to be higher than in many previous studies, the ammonium concentrations after catalysts in raw exhaust were below instrument FTIR detection limit 2 ppm in all studied cases. Therefore, possibly even low ammonia slips from catalysts may have an effect on secondary aerosol formation.

The text was modified to include the significant role of ammonia on nitrate formation.

Pg 12: *The mass concentration of nitrate in secondary particles increased as the catalyst temperature decreased. This could not be explained by catalyst performance improvement: gaseous NOx levels remained similar at all catalyst temperatures or rose as catalyst temperature increased (see Lehtoranta et al., 2016). Because ammonia concentrations after catalyst were low, below 2 ppm in all cases, the effect of catalyst temperature on ammonia emission could not be measured. However, ammonium concentrations measured by SP-AMS correlated rather well with nitrate concentrations. Therefore, we suggest that ammonium increase was related to the nitrate increase. Also the sulfate concentrations could partly explain the behavior of the nitrate concentrations.*

Pg 11 Ln 22. Why was a collection efficiency of 0.5 used? This is clearly suspect given the discrepancy between EEPS and AMS. What, in quantative terms, would be the time dependent collection efficiency using the parametrization by Middlebrook and how would this effect the magnitude of the SOA formation and SOA/POA ratios?

The time-dependent CE was calculated according to the parametrization by Middlebrook et al. (2012). The parametrization changed the total aged concentrations in Figure 2 only little – now the CE is 0.45 while in the first version of the manuscript it was 0.5. The latter terms of the equations [4] and [6] in the paper of Middlebrook et al. were on average 0.22-0.43, leading to composition-dependent collection efficiencies 0.45. In consequence, the choice of the default CE does not explain the discrepancy between EEPS and AMS. The usage of the time-dependent parametrization affected a little the SOA formation and the SOA/POA ratios. By contrast, the parametrization made significant changes in Figure 4 and in Table 3. All figures in the manuscript have now been re-calculated using the parametrization by Middlebrook. The most significant impact of the parametrization was on the evaporation temperatures of sulfate, which were decreased by about 20-40 °C.

Pg 12 Ln 1. The authors extrapolate from one natural gas engine to natural engines in general (see major comments).

Here we did not extrapolate from the NG engine tested in this paper to natural gas engines in general. Instead, we referred to the fact that natural gas engines emit less particle mass than engines fueled with conventional fuels (e.g. Pirjola et al. 2016, Anderson et al. 2015, Prati et al. 2011, Bielaczyc et al. 2014). The text was slightly modified.

Pirjola, L., Dittrich, A., Niemi, J.V., Saarikoski, S., Timonen, H., Kuuluvainen, H., Järvinen, A., Kousa, A., Rönkkö, T., Hillamo, R. Physical and Chemical Characterization of Real-World Particle Number and Mass Emissions from City Buses in Finland (2016) Environmental Science and Technology, 50 (1), pp. 294-304.

Anderson, M., Salo, K. and Fridell, E.: Particle- and Gaseous Emissions from an LNG Powered Ship (2015), Environ. Sci. Technol., 49 (20), pp. 12568–12575.

Prati, M.V., Mariani, A., Torbati, R., Unich, A., Costagliola, M.A., Morrone, B. Emissions and Combustion Behavior of a Bi-Fuel Gasoline and Natural Gas Spark Ignition Engine (2011), SAE International Journal of Fuels and Lubricants, 4 (2), pp. 328-338.

Bielaczyc, P., Woodburn, J. and Szczotka, A.: An assessment of regulated emissions and CO2 emissions from a European light-duty CNG-fueled vehicle in the context of Euro 6 emissions regulations (2014), Appl. Energy, 117, pp. 134–141.

Pg 14: *Natural gas engines emit very little particle mass, which can make them less harmful to human health than corresponding gasoline-, diesel- or marine-fuel-oil-fueled engines.*

In its then state, the intensity of the PAM UV lights could not be varied or changed. For that reason, the authors do not have this data, although we agree with the Referee that it would have provided important information. Our study was, to our knowledge, the first one focusing on secondary aerosol formation from a natural gas engine and we observed that measurable amounts of secondary aerosol forms from NG engine exhaust in a PAM chamber. The research topic suggested by the Referee will be hopefully studied in the future.

-- More importantly, this creates the problem of comparing the data obtained in this study with other chamber and OFR studies (that have quantified secondary particle formation on at least a semi-continuous basis for photochemical age) and presents challenges in making conclusions about the natural gas engine (see next major comment below).

We removed the claims on the advantageousness of natural gas engine in comparison to other engines. The table 2 provides the reader in a short and clear form the information that the photochemical ages simulated in our study differ from many of the previous SOA articles. It also gives the reader an idea of the magnitude of SOA formation potential from engines in general. For that reason, we did not change or modify the representation of the table.

Second, the authors have not considered known artifacts of using an OFR related to the loss of vapors to the OFR walls and accelerated chemistry (Palm et al., 2016) and the short residence times and small condensation sinks (Jathar et al., 2017) in an OFR that might not allow the oxidation products to condense as secondary particles (since the natural gas engine produced very few primary particles, I

think this might be an important factor). All of these contribute to underestimating the secondary particle formation in OFRs, again affecting the comparison with chamber studies. A discussion of these effects and an attempt to correct for these effects will tell the authors if and how these artifacts could affect the results and conclusions. I would refer the authors to Palm et al. (2016) for methods to correct for these artifacts.

As requested, we used the model of Palm et al. (2016) to estimate the magnitude of wall losses and the significance of accelerated chemistry in the PAM. According to the model, the losses of condensable organic oxidation products were approximately 6 % (sticking coefficient $\alpha = 1$) and losses of sulfuric acid were 2 %. Because the average particle size distribution of entering and exiting sample is used in the model, the effect of small condensation sink entering the PAM was not emphasized in the model, although we agree with the Referee that in applications with low primary particle concentration and size, such as a natural gas engine, small condensation sinks can hinder condensation of organic vapors in PAM. Because the modeled losses were low, SOA production factors or concentrations were not corrected for. A new chapter discussing the PAM losses and artifacts was added.

A new chapter concerning particle and precursor losses was added in pages 13-14:

*Organic precursor losses and artifact effects of the accelerated chemistry in the PAM have been evaluated following the method of Palm et al. (2016) for the cases in Figure 2. HRLPI number size distributions were used to calculate the condensation sink needed in the loss calculation. Molar mass of 200 g mol$^{-1}$, diffusion coefficient of $7*10^{-6}$ m$^2$ s$^{-1}$ (Tang et al., 2015) and rate constant for reaction with OH of $1*10^{-11}$ (Ziemann and Atkinson, 2012) were applied. For sticking coefficient selection $\alpha = 1$ (assumed by Palm et al. 2016), the precursor penetration was $0.94 \pm 0.03$, but for $\alpha = 0.1$ the precursor penetration was $0.64 \pm 0.15$.*

*The losses of sulfuric acid were also calculated in the same way. Diffusion coefficient of $1*10^{-5}$ (Hanson and Eisele, 2000), $\alpha$ of 0.65 (Pöschl et al., 1998) and molar mass of 98.079 g mol$^{-1}$ were used for sulfuric acid. The penetration of sulfuric acid was $0.98 \pm 0.01$. According to Lambe et al. (2011), SO2 losses in the PAM are negligible. It can be concluded that the effect of precursor losses and artifacts in the PAM was not substantial in our measurements. The measurement of ammonium nitrate and ammonium sulfate is difficult because ammonia sticks on the walls of sampling systems and instruments (Suarez-Bertoa et al., 2015; Heeb et al., 2012, 2008), which may result in wall losses or an artifact on subsequent measurements. The penetration of ammonia could not be calculated, but the measured ammonium concentrations varied clearly from one case to another, implicating that the source of the ammonia was indeed the exhaust line instead of e.g., the walls of the PAM. However, longer times for stabilization of the SP-AMS concentration would have been advantageous for the reliability of ammonium, and as a consequence, nitrate particle formation.*

*Karjalainen et al. (2016) and Timonen et al. (2017) estimated the effect of particle losses in a similar PAM chamber to be small. The particle losses measured by Karjalainen et al. (2016) depend on particle size and are below 10 % at the particle sizes with most particle mass. An exact calculation of the particle losses in the PAM chamber is not possible because the particle size and number increase while the aerosol sample flows through the chamber. An upper limit for the particle losses in the chamber can be given, calculated using the average of HRLPI particle number size distributions before and after the chamber (similarly to the precursor-loss calculation by Palm et al., 2016) and the PAM particle loss curve. The particle loss according to this examination was $20.3 \pm 3.7$ %. Most probably, however, the actual particle losses in the chamber were smaller because majority of the mass actually was located at larger particle sizes that HRLPI is unable to measure, where particle losses are smaller.*

-- In addition, a brief discussion of the pitfalls in using the OFR (e.g., accelerated chemistry leading to multiple reaction steps in the gas-phase) and particularly its comparison to chamber data would be helpful.

A short discussion was added, mentioning different fates/losses and advantages/disadvantages in comparison to chamber studies.

Pg 13:

**3.4 PAM artifacts and losses**

So called smog chambers are an established method of measuring SOA formation. An oxidation flow chamber such as PAM provides some advantages in comparison to smog chambers, such as higher degree of oxidation, smaller physical size and a short residence time, which allows measurements with higher time resolution. Smog chamber walls may cause also large wall losses and influence the chemistry in the chamber (Bruns et al., 2015). On the other hand, smog chambers may simulate atmospheric oxidation of organic precursors better than oxidation flow chambers due to their more tropospheric oxidant concentrations and longer residence times (Lambe et al., 2011).

PAM method has been designed to produce the maximum potential aerosol mass from precursor gases (Kang et al., 2007). In that stage, the oxidation products of precursors have condensed to particle phase and formed secondary aerosol. However, because the oxidant concentrations are unrealistically high in PAM, the UV light intensity used is non-tropospheric and the residence times are much shorter than in atmosphere (e.g., Simonen et al., 2017), precursor oxidation products have also other possible fates; they can be oxidized too far and form non-condensable oxidation products before condensation (accelerated chemistry) and they can exit the reactor before the condensation occurs. Also, precursor oxidation products can be lost on the PAM walls although the losses on the walls are minimized by the chamber design (Lambe et al., 2011). These other fates than condensing on particle phase are viewed here as PAM artifacts and losses.

2. Comparison with earlier studies: Based on the above comment, I have a few questions that center around the conclusions of this study: (i) can the results of this study at high photochemical ages (several days) be compared to chamber studies of gasoline and diesel vehicles performed over several hours of photochemistry (in my opinion, probably not)?

See reply 2 (iii).

(ii) how would the OFR artifact corrections change the comparison described in Table 2 (the Table could also add recent measurements made by Jathar et al., 2017 for diesel engines)?

Artifact corrections would change the comparison in Table 2 only little, since the SOA precursor losses were small, based on the model of Palm et al. (2016).

The recent article by Jathar et al. (2017) was referred, as well as a recent article by Timonen et al. (2017).

(iii) how confident are the authors in claiming that the secondary particle production from natural gas engines is quite small compared to gasoline and diesel engines, especially in urban areas where photochemical ages are low (0.5-1.5 days).

In this version of the manuscript, we try not to directly compare the secondary particle production of our engine directly to literature with low photochemical ages, but to give the reader an idea of the magnitude, the SOA formation potentials reported in literature are mentioned. The differences of the

photochemical ages are highlighter every time any implication of a comparison is made. The tunnel study of Tkacik et al. (2014) had similar oxidation method and similar photochemical ages as we tested. Therefore, in our opinion, a more direct comparison to that article could be done.

A few sentences stating the advantageousness of natural gas engines were removed and the text around table 2 was defined.

Pg 9: *Table 2 also contains SOA production factors of secondary organic aerosol for different diesel and gasoline vehicles obtained from the literature. Although the total aged particulate matter production of the investigated NG engine was much larger than its primary particle emissions, it was smaller than SOA production from in-use diesel and gasoline vehicles in the literature (Tkacik et al., 2014). The SOA formation potential from the NG engine, measured by SP-AMS, was similar to that of a diesel vehicle equipped with a catalytic converter or to that of a hot start gasoline vehicle. On the other hand, the photochemical age that was simulated by a chamber in the different studies varied greatly. This is why the comparison of the SOA production factors should be done very carefully, if at all. The longest atmospheric ages in the literature collected in Table 2 were achieved in our study.*

Pg 15: *To estimate the quantity of the NG engine exhaust's SOA formation potential, it was on the same level as or lower than the SOA formation potential of a diesel vehicle equipped with an oxidation catalyst or that of warm (hot start) gasoline vehicles. However, the photochemical age that was produced by the PAM chamber in our study was longer (several days) than the photochemical ages achieved in the previous studies (several hours). Therefore, the SOA formation potential must not be directly compared.*

(ii) how relevant are the high photochemical age results in this study to the atmosphere given that the there are other processes (e.g., transport, deposition) that are relevant at the same time scale?

The purpose of the potential aerosol mass (PAM) chamber method is to try to achieve the maximum potential formation of secondary aerosol mass. Unfortunately, presumably, the maximum was not quite achieved for example because of the losses in the oxidation flow reactor and because we could not adjust the achieved photochemical age in the chamber. This, however, was the aim of the study. Deposition and transport are important processes when the effects of emissions are evaluated. However, the evaluation of those processes was not the aim of this study, but instead the aim was to evaluate the maximum potential of natural gas engine exhaust to form secondary aerosol mass.

The photochemical ages that were simulated by the PAM chamber were 4-11 days. The tropospheric lifetimes of many aerosol species has been evaluated to be of the same time scale (IPCC 2013). We admit that shorter photochemical ages would have been also relevant to the atmosphere and because of transportation, shorter photochemical ages would have been more relevant to highly populated areas. However, this does not mean that the age of approximately one week is irrelevant.

IPCC (2013), Climate Change 2013: The Physical Science Basis, Cambridge Univ. Press, Cambridge, UK.

The operation of the OFR and the artifacts linked to the OFR do not allow for definitive answers to any of the above questions. This fact needs to be considered for the summary/conclusions from this work.

A sentence stating this fact was added.

Pg 15: *Also, despite the attempts to model PAM related losses and artifacts and to estimate particle losses in PAM, the measurements performed with PAM still involve uncertainties.*

3. Style and phrasing: The writing, while detailed, needs to be paid more attention and the style and phrasing need to be improved throughout the manuscript. Here are just a few examples: (i) Page 11,

line 30: 'Although' instead of 'However, although', (ii) Page 11, line 8: 'remained similar at all catalyst temperatures' instead of 'remained similar in all catalyst temperatures', (iii) Page 10, line 29: 'waited' instead of 'was waited', (iv) Page 7, line 6: 'condense' instead of 'condensate'.

The mentioned examples were corrected. In addition, the manuscript was proof-read by a native speaker. As a result, numerous small improvements in the style and phrasing were made in the manuscript, practically in every page.

Minor comments:

1. Page 2, line 27: Is there a reference for the comment on diesel vehicles? Also, can this comment be made for other combustion/energy sources?

The sentence was modified to better serve its purpose. The comment on diesel vehicles referred to the particulate number regulation that concerns diesel vehicles (nowadays also gasoline vehicles) and is considered as more stringent than the particulate mass regulation.

Pg 2: *Particle number and mass emission regulations for passenger cars and heavy-duty engines have substantially decreased the primary particle emissions from vehicles (e.g., May et al., 2014; Johnson, 2009). Secondary particle precursor emissions or secondary aerosol formation potential are not directly regulated, but some of the current emission regulations affect secondary particle precursor emissions indirectly. For instance, oxidative catalysts reduce the total hydrocarbon emissions and thus probably the emissions of secondary organic aerosol precursors; simultaneously, they also change the oxidation state of inorganic compounds. Furthermore, the mandatory national targets of 10 % biofuel in gasoline in EU may have decreased the SOA formation in the atmosphere (Timonen et al., 2017).*

2. Page 2, line 30: May be use the word 'oxidative catalyst' to refer to the aftertreatment device?

Corrected as requested.

Pg 2: *For instance, oxidative catalysts reduce the total hydrocarbon emissions and thus probably the emissions of secondary organic aerosol precursors; simultaneously, they also change the oxidation state of inorganic compounds.*

Page 3, line 4: Is there a newer reference instead of Weitkamp for the last sentence? Examples: Gordon et al., 2014 or Jathar et al., 2017.

Also Gordon et al and Jathar et al were referred to and the sentence was slightly modified.

Pg 3: *From diesel vehicles without a particle filter, the SOA mass formation potential is of the same magnitude as or lower than the primary particle mass emission (Jathar et al., 2017; Gordon et al., 2014b; Weitkamp et al., 2007).*

4. Page 4, lines 9-14: It would help if there was a brief mention of the size, horsepower, typical application, on- or off-road characteristics for the natural gas engine used.

The first sentence was modified to mention that the engine was a gasoline passenger car engine that was retrofitted to run with natural gas.

Pg 4: *The engine was a retrofitted passenger car spark ignition engine using Russian pipeline natural gas as fuel.*

Pg 4: *The two operation modes simulated typical natural gas power plant operation with the retrofitted passenger car engine and thus did not directly represent real-world driving.*

5. Page 4, lines 15-24: What is the rationale for using two different aftertreatment configurations and the different temperatures? Is it based on the drive cycle expected for these engines? It is likely this fact is hidden in some of the earlier literature by the group but it would be beneficial to mention it here.

Catalyst performance depends on the temperature of the catalyst or of the exhaust that flows through it. In our measurement, behavior of the catalysts in lower and higher temperatures were tested in order to define the effects of changing exhaust temperature on emissions and to find the best operation conditions for the tested catalysts. Regarding this paper, especially the effect of catalyst temperature on the formation and characteristics of primary and total aged particulate matter could be studied.

Pg 4: *Catalyst performance depends on the exhaust temperature (e.g. Lehtoranta et al., 2016). By using the catalysts at different temperatures, the effect of catalyst temperature on the formation and characteristics of primary and total aged particulate matter could be studied.*

6. Page 5, lines 9-29: This is a novel application of the MCM to determine photochemical exposure given that exhaust reactivity can significantly suppress OH exposure. Peng et al. (2015) and Li et al. (2015) have developed empirical relationships to calculate OH exposure based on the mixture reactivity, residence time, and humidity in OFRs. Can the authors comment on why the method of Peng+Li was not used to calculate OH exposure? Or in other words, is there reason to believe that the MCM offers better predictions? Has the MCM output been evaluated against OH exposure measurements for an OFR application?

The model by Peng+Li was not available and a similar model offered by PAM User Manual uses Euler method and is not very precise. Furthermore, the model is not flexible and lacks the possibility to add reactions. The combination of MCM and KPP offers a flexible method with good solver options for modeling gas-phase chemistry. MCM is widely used and proven to work. The free parameters of the model were obtained using $SO_2$ measurements. The predicted OH exposures agree well with those derived from $SO_2$ reduction in PAM chamber.

Pg 5: *The model has been tested against $SO_2$ reduction measurements in the PAM chamber.*

7. Page 5, lines 30-34: I am not sure I understand how the RH in the PAM chamber was as high as 80% when the clean dilution air (which was dry or 30%) accounted for 24 parts by volume to the exhaust. The mass balance on the water vapor does not seem to work out. The only way to explain the result is that water vapor is being generated in the PAM?

The dilution ratio prior to PAM was only 6. ("*The PAM chamber was placed between the two dilution stages*"). That is why the high RH was possible. The dilution air was dry. To clarify the sampling and dilution system, the Instrumentation and data analysis chapter, first paragraph was slightly modified.

*For the particle measurement instruments, the sampling system consisted of a porous tube diluter (PTD, Mikkanen and Moisio 2001; Ntziachristos et al. 2004 with dilution ratio (DR) 6, followed by a residence time chamber with a residence time 6 s. The dilution air was heated to 30 ◦ C in order to achieve constant dilution conditions. A second dilution stage was carried out with an ejector diluter (Dekati Ltd.) with DR 4.*

8. Page 6, lines 27-31: Subtraction of primary particle mass needs to be done when presenting the secondary particle production factors. For instance, SOA emission factors in Table 2.

In fact, in Table 2 this was already done. This has been mentioned in the text: *"To be able to compare the SOA production factors, here primary organic aerosol has been subtracted from the total aged organic aerosol."* To prevent confusion, the mention was added also in the caption of the Table 2 and in the first paragraph or chapter 3.1.

Table 2: *Primary organic aerosol has been subtracted from the total aged organic aerosol.*

Pg 7: *To enable the comparison to literature, an exception is made when presenting secondary particle production factors.*

Do the authors know of the recent work of Link et al. (2017) where they examined photochemical production of inorganic compounds from light-duty vehicle exhaust? That might be another study to incorporate within the inorganic particle production discussion.

The following insertions or modifications were made in the results and conclusions sections.

Pg 8: *Link et al. (2017) found that even high NOx emissions can produce negligible amounts of secondary nitrate aerosol if related ammonia emissions are small. Because secondary ammonium nitrate aerosol formation is limited by ammonia, its formation is probably more related to the exhaust after-treatment than the fuel. The exact ammonia concentrations in the raw exhaust cannot be given because they were below the instrument detection limit 2 ppm. According to these measurements, also low ammonia emissions may have atmospheric importance as secondary inorganic aerosol precursors.*

Pg 15: *Therefore, what limits the nitrate mass in particles is most likely the availability of ammonia which is more related to the exhaust after-treatment than fuel or combustion process.*

9. Page 7, lines 17-21: Long sentences beginning with 'It is possible' and ending with 'into the atmosphere'. Consider breaking into multiple sentences.

The sentences were slightly modified in order to make them more readable.

Pg 8: *It is possible that if the catalyst conditions are favorable, the particulate matter that would otherwise condense on particles in PAM chamber condenses on particle phase already in the cooling and dilution process. I.e. if the catalyst sufficiently oxidizes the exhaust gases thus lowering their saturation vapor pressure, they condense or nucleate already when released from the tailpipe and not later on in the atmosphere.*

10. Page 8, lines 14-18: Why isn't the standard calculation of EF based on the CO2 concentrations and carbon intensity of the fuel, employed in this work?

Thank you for pointing this out. A small mistake was spotted because of the comment. The calculation actually was performed correspondingly to the "standard calculation" of emission factor in unit mg $kg^{-1}_{fuel}$. The carbon intensity, calculated from fuel composition, was 0.74 $kg_C$ $kg_{fuel}^{-1}$. The equation used by e.g. Gordon et al. (2014) can be reformulated as follows: [P] stands for the concentration of particles, $C_f$ stands for carbon content of the fuel and $x_{co2}$ for the molar fraction of $CO_2$.

$$EF = \frac{[P]}{[CO_2]} \frac{M_{CO_2}}{M_C} C_f = \frac{[P]}{x_{CO_2}} \frac{RT}{pM_{CO_2}} EF_{CO_2}$$

Calculated from the fuel composition information, the $EF_{CO2}$ was 2730 $g_{CO2}$ $kg^{-1}_{fuel}$. However, mistakenly, $EF_{CO2}$ 2540 g $_{CO2}$ $kg^{-1}_{fuel}$ was used previously (Huss et al. 2013). Now, the calculated $EF_{CO2}$ 2730 g $_{CO2}$ $kg^{-1}_{fuel}$ has been applied e.g. in the figures in Table 2. The procedure of emission factor calculation is now described in more detail in the text.

Huss, A.; Maas, H.; Hass, H. Well-to-wheels analysis of future automotive fuels and powertrains in the European context. Tank-To- Wheels (TTW) Report Version 4.0, July 2013. http://iet.jrc.ec.europa. eu/about-jec.

Gordon, T. D., Presto, A. A., Nguyen, N. T., Robertson, W. H., Na, K., Sahay, K. N., Zhang, M., Maddox, C., Rieger, P., Chattopadhyay, S., Maldonado, H., Maricq, M. M. and Robinson, A. L.: Secondary organic aerosol production from diesel vehicle exhaust: Impact of aftertreatment, fuel chemistry and driving cycle, Atmos. Chem. Phys., 14(9), 4643–4659, doi:10.5194/acp-14-4643-2014, 2014.

*Pg 7: Emission factors were calculated from fuel composition and engine performance information. Residual $O_2$ in exhaust was 6.2-6.3 %, the power of the engine was 12 kW and 20 kW and the combustion air flow into the engine were approximately 100 and 115 kg h$^{-1}$ at the engine modes 1 and 2, respectively. Calculated from the fuel composition information, the emission factor for $CO_2$ $EF_{CO2}$ was 2730 g kg$_{fuel}^{-1}$ and carbon intensity 0.74 kg$_C$ kg$_{fuel}^{-1}$*

*Pg 9: Emission factors or secondary aerosol production factors in different units can be calculated from the presented particle mass concentrations with use of following factors. If a unit factor mg kg$^{-1}$ fuel is needed, a factor of ca. 22 m$^3$ kg$^{-1}$ fuel can be applied to multiply the particle concentration (Calculation e.g. in Jathar et al., 2017; Gordon et al., 2014b). In order to obtain emission and production factors in unit kWh$^{-1}$, a factor 2.7 m$^3$ kWh$^{-1}$ (Mode 1) or 4 m$^3$ kWh$^{-1}$ (Mode 2) can be similarly used. These factors are derived from the fuel composition and engine performance information provided in Sect. 2.1 and 2.2, and exhaust CO2 concentration.*

11. Page 9, line 21: While I understand what the authors mean, 'oxidated alpha-pinene particles' is not correct.

Corrected in form:

*Pg 10: An et al. (2007) measured the volatility of secondary organic aerosol produced during alpha-pinene photo oxidation --*

12. Figure 2: The pie charts are too small. Please consider increasing them in size. Will the plots a, c, and d benefit from being in emission factor units?

The size of the pie charts has now been increased. In our opinion, the plots 2a, 2c and 2d do not benefit from being in emission factor units. The values in the plots can be easily transformed to emission factor units by a reader, if needed, by using the factors that are given in the text. We also believe that for many readers, concentrations are more intuitive and understandable than emission factors.

13. Figures 5 and 6: Do the EEPS and HRLPI size distributions compare well with the AMS size distributions? Can the EEPS, HRLPI, and the AMS data be combined to provide a more complete picture of the size- and composition-resolved emission/production of particles?

The instruments measure different size ranges, as seen in the figure below. The two instruments that measure the aerodynamic diameter of the particles seem to compare quite well with each other between 47-124 nm in a loglog plot but in the two lower right-hand-side plots there is a difference between the concentrations. EEPS particle size mode is always below 80 nm whereas both HRLPI and SP-AMS imply that the mass size distribution peak lies well above 100 nm. This is at least partly due to the different measurement principle of EEPS. Both instruments HRLPI and EEPS miss the second peak above 300 nm and SP-AMS cannot give any information on the particles with diameter below about 40 nm. The figure below has now been added in the supplementary in case the future readers have the same question in mind. A modification in the manuscript was also done.

[Figure]

Figure S3 Mass size distributions measured by SP-AMS (sum of the size distributions of organics, sulfate, nitrate and ammonium), EEPS and HRLPI. Note: on the x-axis there is aerodynamic particle diameter for HRLPI, vacuum aerodynamic diameter for SP-AMS and mobility diameter for EEPS.

Pg 13: *The mode with smaller particle size was dominated by organics.*

Pg 13: *The best overall picture of is therefore gained with a combination of SP-AMS and HRLPI. See supplement for a comparison of the size distributions measured by different instruments in a same figure. The two instruments that measure the aerodynamic diameter of the particles (HRLPI and SP-AMS) compare quite well with each other in the size range 47-124 nm.*

Pg 13: *EEPS also underestimated the mass of particles with diameter above 200 nm (see supplement).*

14. Can the authors speculate on the species in the exhaust and photochemical mechanisms that can explain the secondary organic particle production?

[revised manuscript text omitted]

*S3 Particle mass size distributions measured with SP-AMS, HRLPI and EEPS. Note that $d_p$ stands for aerodynamic diameter for HRLPI, vacuum aerodynamic diameter for SP-AMS and mobility diameter for EEPS.*

---

## Author Response (AR2)

Dear Editor and the Referees,

A few improvements in the language of the manuscript were done as requested. The corrections were done in the first paragraph of the section 2.2, in the sections 3.3 and 3.4 and in the Conclusions section. Also, one reference (Mikkanen et al. 2001) was corrected.

Yours sincerely,

Jenni Alanen and the co-authors